# Bayesian-Informed Diverse Sampling for Calibration of Fine-Tuned Foundation Models with Evidential Ensembles

## Abstract

As large foundation models become increasingly crucial in critical domains, concerns regarding the handling of incorrect content from these models continue to grow. Fine-tuning these models for downstream tasks exacerbates the risks of generating poor-quality outputs, as potential domain shifts and variability in downstream task data size can significantly impact performance. Recent efforts have focused on improving model reliability by quantifying uncertainty in generated content. State-of-the-art solutions generate diverse answer paths through probabilistic decoding methods like top-$k$, nucleus sampling, or beam search, using the inconsistencies among these answers as a measure of uncertainty. From a Bayesian perspective, aggregating these diverse paths effectively marginalizes over the approximate model parameter distribution, while a single output based on maximum conditional probabilities serves as a point estimate of that distribution. This further justifies the superiority of multi-output methods over single-output approaches. However, fine-tuning on limited downstream data can result in a posterior distribution that is more complex, flatter, and potentially multimodal, rendering current methods insufficient for accurate uncertainty quantification. To better approximate this complex posterior, we propose a novel diversity-inducing ensemble approach guided by Distributionally Robust Optimization (DRO) and evidential theories. Our method is theoretically guaranteed to improve calibration performance. Empirical results across benchmark datasets and large visual language models demonstrate the effectiveness of our approach in estimating content quality and enhancing model reliability.

## 1 Introduction

Large foundation models, such as Large Language Models (LLMs) (*e.g.,* Llama (Meta, 2024), Gemma (Google, 2024), Vicuna (Team, 2023)), and large visual language models (LVLMs) (*e.g.,* LLaVA (Liu et al., 2024a), Blip2 (Li et al., 2023a), Flamingo Alayrac & Donahue (2022)) have achieved remarkable success a wide range of natural language generation (NLG) tasks, including visual question answering, text summarization, image captioning, and image-text retrieval (Brown, 2020; Wang et al., 2022a). As foundation models are increasingly deployed in diverse fields, concerns about content quality have continued to grow, particularly in high-stakes domains such as education and medicine. At the same time, the pretrain–then–fine-tune paradigm has become the dominant approach for adapting foundation models to downstream applications via few-shot fine-tuning. However, given potential domain shifts and variability in the size and quality of downstream training data, fine-tuned models are especially susceptible to generating inaccurate or misleading outputs, amplifying risks in critical settings.

To tackle this critical challenge of ensuring content quality, recent work has explored detecting incorrect outputs by quantifying model uncertainty/confidence. A common strategy is to estimate confidence using the conditional probabilities of generated tokens, which capture uncertainty in next-token prediction conditioned on prior context (Ren et al., 2022). However, paraphrased variants of the same semantic content can yield different token-level probabilities, which limits it reliability. To mitigate this, several approaches prompt models to generate an explicit self-assessment of confidence (*a.k.a.* verbalized confidence) over the entire generated output (Xiong et al., 2023; Wang et al., 2024;

Lin et al., 2022a). However, empirical studies reveal a tendency toward systematic overconfidence, leading to unreliable detection accuracy (Xiong et al., 2023). Another direction involves training a linear probe on embedding layers to predict uncertainty scores, using labeled correct and incorrect samples for supervision (Li et al., 2023b; Azaria & Mitchell, 2023; Marks & Tegmark, 2023). However, this method is highly sensitive to the choice of the embedding layer. Its effectiveness also hinges on the size and quality of the training data and overfitting in the probe further will compromise the reliability of confidence estimation.

All of the above approaches for uncertainty quantification of generative models rely on a single generated output. In contrast, generating multiple outputs promotes diverse response pathways to the same query (Bakman et al., 2024; Nikitin et al., 2024; Farquhar et al., 2024; Kuhn et al., 2023b). In this context, obtaining the final answer can be seen as performing marginalization across the various answer paths (Wang et al., 2022b). The inconsistency among these diverse answers can then be exploited to quantify the overall uncertainty (Chen et al., 2024; Kuhn et al., 2023a; Lin et al., 2023; Bakman et al., 2024; Nikitin et al., 2024; Farquhar et al., 2024). Multiple outputs can be generated through various decoding methods, such as top-$k$ sampling, nucleus sampling, multinomial beam search, and temperature scaling. Moreover, recent research has examined methods for merging semantically equivalent answers into a single plausible output (Farquhar et al., 2024; Kuhn et al., 2023b; Nikitin et al., 2024; Bakman et al., 2024).

Multi-output approaches have been shown to improve empirical performance in prior studies Farquhar et al. (2024); Kuhn et al. (2023b); Xiong et al. (2023). However, they become insufficient to accurately quantify the model uncertainty after fine-tuning. When a pre-trained foundation model is adapted to a downstream task, potential domain shifts and the limited size and noise of task-specific data can lead to poor calibration due to overfitting. Consequently, the uncertainty estimates produced by standard multi-output methods become unreliable. This phenomenon can be more precisely understood from a Bayesian perspective. Consider a pre-trained model parameterized by $\theta$. Using a probabilistic decoding mechanism to generate multiple outputs can be interpreted as approximating the distribution $p(\theta)$ over model parameters through sampling. Aggregating the diverse answer paths obtained in this way effectively marginalizes over the approximate parameter distribution, producing the predictive distribution of the final output: $p(y) = \int_\theta p(y|\theta)p(\theta)\mathrm{d}\theta$. This explains why multi-output methods are fundamentally more informative than generating a single output via the maximum joint conditional likelihood, which corresponds to a point estimate of the parameter distribution and thus cannot capture uncertainty in the model parameters. For foundation models trained on massive datasets, $p(\theta)$ is typically sharply peaked, making decoder-based sampling sufficient to approximate the true distribution. After fine-tuning on a downstream dataset $D$ using standard parameter-efficient fine-tuning (PEFT) methods (*e.g.,* LoRA), the model weights are updated to $\theta + \Delta\theta$. The posterior distribution $p(\theta|D)$ can be much flatter and multimodal due to the small size and noise in the task data. Consequently, decoder-based sampling provides a limited approximation of the true posterior.

Ensemble-based methods have been explored to better approximate this posterior (Malinin & Gales, 2020), typically by randomly initializing ensemble members to produce diverse parameter sets. While this improves over sampling from a single model, random initialization alone often fails to generate sufficient diversity to adequately explore the complex posterior distribution. To address this gap, we propose a novel Bayesian-informed, diversity-inducing ensemble that more accurately approximates the posterior distribution of model parameters, leading to improved uncertainty quantification for detecting incorrect outputs. Our ensemble approach is grounded in Bayesian modeling and guided by the principles of Distributionally Robust Optimization (DRO) and evidential theory. For supervised fine-tuning, we introduce a loss function designed to maximize the joint probability of supervision tokens. In our formulation, each ensemble component is fine-tuned to focus on different levels of sample difficulty. By controlling the parameters in the loss function as derived from the DRO principle, we enable the model to emphasize varying levels of difficulty, thereby ensuring a diversity-driven exploration of the complex posterior distribution of model parameters. We theoretically show that the joint probability can be decomposed into two distinct sources of evidential uncertainty: confusion in prediction and lack of knowledge. This decomposition allows our ensemble to adaptively focus on different difficulty levels defined by fine-tuned evidential uncertainty. Our contributions are threefold:

- a novel DRO-evidential loss function to promote diversity among ensemble components for accurate approximation of the posterior distribution of model parameters,
- a theoretical analysis demonstrating the calibration improvements guaranteed by this loss function,

- empirical validation of the method's effectiveness across multiple benchmark datasets and large visual language models.

## 2 RELATED WORK

**Uncertainty Quantification in LLM.** With the widespread applicability of the LLMs, there has been an increasing interest in understanding and quantifying the uncertainty of these models (Ling et al., 2024; Xiao & Wang, 2021; 2018; Lin et al., 2024). A straightforward way is to utilize the human-like language generation and ask the model to answer the question with a confidence measure. This way of eliciting confidence is termed verbalized confidence (Yang et al., 2024; Tao et al., 2024; Lin et al., 2022b; Wang et al., 2024). Although convenient to use, it is observed that verbalized confidence tends to be overconfident, and the performance is highly sensitive to the use of prompt (Xiong et al., 2023). Another naive baseline is to leverage conditional probabilities of the generated tokens (Ren et al., 2022; Bakman et al., 2024). Some works also train a linear probe using an embedding layer as input and truthfulness measure as the target (Kapoor et al., 2024; Azaria & Mitchell, 2023; Liu et al., 2024b). A training dataset with true/False labels is required to supervise the training process (Azaria & Mitchell, 2023). Some other works train an auxiliary model (encoder model) to output the confidence based on confidence labels (Ulmer et al., 2024) instead of the linear probe. Recent state-of-the art methods sample multiple output from the model and quantify consistencies between the answers as the confidence. Inconsistencies between tokens are considered to output uncertainties in generations (Fomicheva et al., 2020; Lin et al., 2023; Farquhar et al., 2024; Kuhn et al., 2023a). Semantic entropy (Kuhn et al., 2023a) extended the concept of entropy at the semantic level by gathering semantically similar concepts and combining their likelihoods before computing the overall entropy. Similarly, (Farquhar et al., 2024) leverages the idea of semantic entropy to detect hallucinations in large language models. Ambiguity among generated content is considered while quantifying inconsistencies between contents via kernel entropy (Nikitin et al., 2024). In this work, we focus on obtaining diverse paths to answer the question such that the complex posterior distribution is approximated better. Ensemble components, when randomly initialized, offer a way to induce diversity in generated paths (Malinin & Gales, 2020). However, we observe limited improvement in the calibration performance with randomly initialized ensemble components. Hence, our novel solution is guided by distributionally robust optimization to induce diversity among the ensemble components.

**Model Calibration.** Model calibration is widely studied in classification problems using the metric of ECE (Naeini et al., 2014; Nixon et al., 2019) or variants of ECE in the form of Maximum Calibration Error (Naeini et al., 2015), classwise ECE (Kull et al., 2017), or adaptive ECE (Nixon et al., 2019). To calibrate the models, several post-hoc mechanisms have been proposed that adjust either logits or probabilities Platt et al. (1999); Mozafari et al. (2018); Pandey et al. (2024); Guo et al. (2017). Worst violation of calibration among partitions of the prediction space is leveraged as a corrective signal to update predictions as a post-hoc recalibration mechanism (Zhao et al., 2021). To mitigate overconfident predictions, an implicit regularization technique, such as mixup (Thulasidasan et al., 2019), label smoothing (Müller et al., 2019), and focal loss (Lin et al., 2017), is proposed, which improves the calibration of the model. Ensemble-based methods (Lakshminarayanan et al., 2017; Sapkota et al., 2023) have been found effective to improve calibration performance. Gradients of the ensemble model's output with respect to the input data are pushed away from each other to boost calibration from ensemble components (Trinh et al., 2023). Some studies extend the study of calibration to large vision models (Guo et al., 2017; Pandey et al., 2024). Tu et al. (2023) study the calibration behavior of the CLIP model, a vision language model. In this work, we study the calibration behavior of large vision language models with auto-regressive generative content as output. Some works in literature study the calibration behavior of natural language generation (Tian et al., 2023; Wang et al., 2024; Ulmer et al., 2024; Huang et al., 2024). In this work, we propose a diversity-inducing ensemble to achieve better calibration of LLMs.

## 3 METHODOLOGY

For the generative setting, we consider a general case and define an input query containing both text and image $x_{\text{query}} = [x_{\text{image}}, x_{\text{text}}]$. Given $x_{\text{query}}$, the tokens generated by the model is $\mathbf{y} = [y_1, y_2, ...y_T]$. The selection of the $t^{th}$ token is based on the probability $p(y_t|y_{<t}, x_{\text{query}})$ conditioned

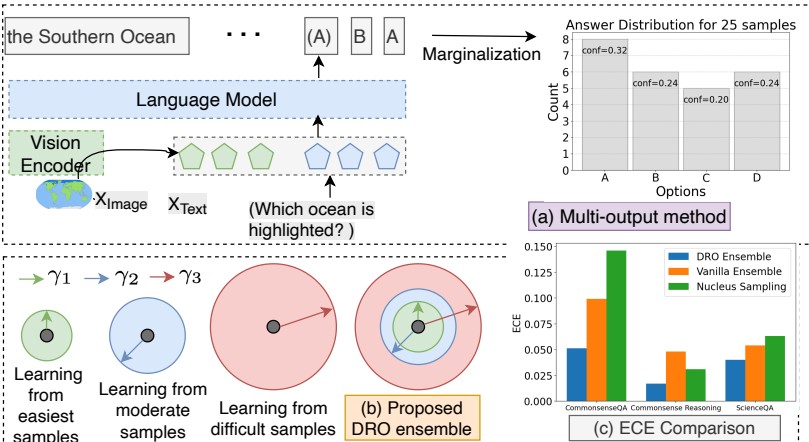

Figure 1: a) Multi-output method: Generate multiple answers for the same query using sampling, and marginalize answers to obtain the most consistent answer sets. b) Proposed DRO ensemble: We leverage the proposed Evidential Uncertainty Guided DRO Loss Ensemble to determine the optimal gradient update that balances easy and difficult samples based on uncertainty estimation. (c) The proposed DRO ensemble achieves better ECE over the multi-output method (nucleus sampling) and vanilla ensemble.

on the input query and previously generated tokens. The goal of uncertainty quantification is to provide an uncertainty score $s$ for the generated output $\hat{\mathbf{y}}$.

### 3.1 DIVERSITY-INDUCING ENSEMBLE VIA DISTRIBUTIONALLY ROBUST OPTIMIZATION

To address miscalibration caused by overfitting during fine-tuning on downstream task data, a common approach is the vanilla ensemble: obtaining $N$ outputs from multiple models fine-tuned on the same dataset with different random initializations. While this approach introduces some diversity among ensemble components, it may still suffer from significant redundancy. To systematically enhance diversity, we propose an ensemble guided by Distributionally Robust Optimization (DRO) Sapkota et al. (2023). We define the DRO loss:

$$\mathcal{L}^{robust} = -\sum_{i=1}^{n} w_i \sum_{t=1}^{T} \log P(y_t | x_{\text{query}}, \boldsymbol{y}_{<t}) = \sum_{i=1}^{n} w_i \mathcal{L}^{\text{ERM}} \tag{1}$$

where $\mathcal{L}^{\text{ERM}}$ is the standard auto-regressive loss and $w_i$ is a weight of $i^{th}$ sample determined by a difficulty score, controlled by a parameter $\gamma$. The exact form of $w_i$ is presented in Equation 4. When $\gamma = 0$ and $w_i = 1$, $\mathcal{L}^{\text{ERM}}$ reduces to the standard ERM loss, assigning equal weight to all samples. As $\gamma$ increases, more emphasis is placed on samples with higher difficulty scores, shifting the model's focus from easy to moderate to difficult samples (illustrated in Figure 1b). By training ensemble components with a different $\gamma$, the model focuses on different levels of sample difficulty, yielding a diverse ensemble. Here, diversity arises from the varied emphasis across training samples of differing difficulty, rather than solely from random initialization.

### 3.2 EVIDENTIAL UNCERTAINTY GUIDED LOSS ENSEMBLE

We derive the form of $w_i$ based on the fine-grained composition of sample difficulty guided by evidential theory.

**Theory of Evidence:** For a $K$ class classification problem, the theory of evidence allows the assignment of belief mass $b_k, \forall k \in \{1, 2...K\}, b_k \geq 0$, along with the uncertainty mass $u, u \geq 0$ Jøsang (2016); Sensoy et al. (2018). They follow the relationship of $\sum_{k=1}^{K} b_k + u = 1$. The model expresses a high value of uncertainty when all the other belief masses are low. Similarly, the model expresses confusion between multiple labels through the sum of belief masses of non-ground truth labels. The fine-grained evidential uncertainties refer to the lack of knowledge ($u$) and confusion between labels ($b^{inc}$). For a model $f$ parameterized with $\theta$, the belief masses and fine-grained

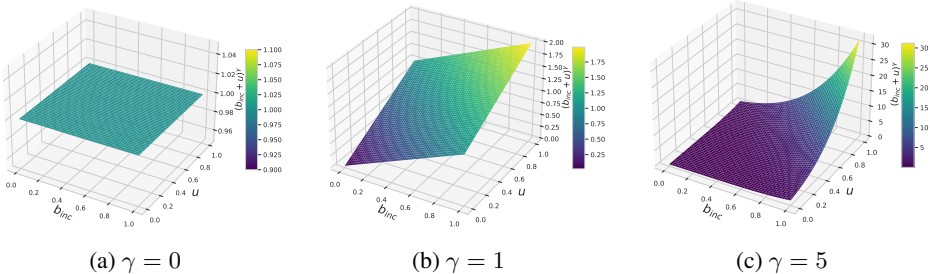

(a) $\gamma = 0$  (b) $\gamma = 1$  (c) $\gamma = 5$

Figure 2: Visualization of weight of the loss with change of $b_{inc}$ and $u$ for different values of $\gamma$.

uncertainties of an input $x$ are calculated with the help of evidence $e_k$:

$$b_k = \frac{e_k}{\sum_{k=1}^{K} e_k + K}, \quad u = \frac{K}{\sum_{k=1}^{K} e_k + K}, \quad e_k = \mathcal{E}(f_\theta(y_k|x)), \quad b^{inc} = \sum_{k \neq gt} b_k \quad (2)$$

**Evidential theory guided difficulty weight:** To calculate uncertainty for output $\boldsymbol{y}$ of training data sample query, we denote logits of $t^{th}$ token as $f_\theta(\boldsymbol{y}_t|x_{\text{query}}, \boldsymbol{y}_{<t})$, which is of size $V$. We leverage fine-grained evidential uncertainties ($u$ and $b^{inc}$) as guides to fine-tune the ensemble component. In the generative context, we define evidence associated with $v^{th}$ token in $t^{th}$ generation step as $\boldsymbol{e}_{tv} = \mathcal{E}(f_\theta(\boldsymbol{y}_{tv}|x_{\text{query}}, \boldsymbol{y}_{<t}))$. The corresponding uncertainty $u_t$ and $b_t^{inc}$ for the $t^{th}$ token are:

$$b_{tv} = \frac{\boldsymbol{e}_{tv}}{V' + \sum_{v=1}^{V'} \boldsymbol{e}_{tv}}, \quad u_t = \frac{V'}{V' + \sum_{v=1}^{V'} \boldsymbol{e}_{tv}}, \quad b_t^{inc} = \sum_{v \neq y_t} b_{tv} \quad (3)$$

where $v$ index represents the tokens among the vocabulary size $V$ used by the model for the $t^{th}$ generation step. We use $V' < V$ tokens in the uncertainty calculation in equation 3 to prevent the model from assigning high uncertainties to all tokens due to the large vocabulary size. To represent the uncertainty of the training sample, we take the maximum of the uncertainty of all the token generation, $u = \max_{t=1}^{T} u_t$, and the maximum of the incorrect belief score among all the token generation, $b^{inc} = \max_{t=1}^{T} b_t^{inc}$. Instead of taking the uncertainty across all tokens, we define difficulty based on the most uncertain token, as the model faces difficulty in learning the key token that plays an important role in the whole content generation. We define the weight of the $i^{th}$ training sample as a function of the defined uncertainty scores of the training sample $i$, resulting in DRO loss as:

$$w_i = (u + b^{inc})^\gamma, \quad \mathcal{L}^{robust} = \sum_{i=1}^{n} w_i \mathcal{L}^{\text{ERM}} \quad (4)$$

The effect of different $\gamma$ values for fine-grained evidential uncertainties is illustrated in Figure 2. When we finetune ensemble components with loss function $\mathcal{L}^{robust}$, each with a different $\gamma$, we obtain $M$ ensemble components. We leverage these components to obtain $N$ generated contents $\mathcal{Y}$. The final answer and confidence are obtained using the following equation:

$$p(\tilde{\mathbf{y}}_k|x_{\text{query}}) \approx \sum_{\mathbf{y} \in \mathcal{Y}} \frac{|\mathbf{y} \equiv \tilde{\mathbf{y}}_k|}{|\mathcal{Y}|}, \quad \hat{\mathbf{y}} = \arg\max_{k \in K}(p(\tilde{\mathbf{y}}_k|x_{\text{query}})), \quad s = \max_{k \in K}(p(\tilde{\mathbf{y}}_k|x_{\text{query}})) \quad (5)$$

Here, $N$ output answers in answer set $\mathcal{Y}$ can be aggregated to $K$ unique answers $\tilde{\mathcal{Y}} = \{\tilde{\mathbf{y}}_k\}_{k=1}^{K}$. The probability distribution over $k^{th}$ unique label is represented by $p(\tilde{\mathbf{y}}_k|x_{\text{query}})$. It is defined as the fraction of the counts of answers that are semantically equivalent to the $\tilde{\mathbf{y}}_k$ out of all the answers in $\mathcal{Y}$.

### 3.3 THEORETICAL ANALYSIS

In this section, we theoretically show how evidential uncertainty-guided loss ensemble results in improved calibration. Specifically, our analysis consists of two parts: (a) fine-grained focal loss weight can be considered the decomposition of the vacuity($u$) and the incorrect belief ($b^{inc}$), and (b) the proposed ensemble loss is upper-bounded by the sum of the cross-entropy, negative of the entropy,

and an additional normalization constant. Because of the negative entropy, minimization of the loss leads to minimization of the cross-entropy loss but prevents wrong samples from taking very high prediction scores because of the entropy maximization regularization term. The lemma below first shows that the focal loss weight can be decomposed as the sum of the vacuity and incorrect belief.

**Lemma 3.1** *Let $w_i = (1 - \hat{p}_{iv})^\gamma$ be the weight associated with the focal loss with $\hat{p}_{iv}$ being the expected probability for the ground truth token $v$ for the $i^{th}$ sample, and $\gamma$ being the hyperparameter controlling the emphasis given to the difficult samples. Then, we can decompose the focal loss as a combination of the vacuity and incorrect belief. Mathematically,*

$$w_i = (\hat{u}_i + b_i^{inc})^\gamma \tag{6}$$

*where $\hat{u}_i = u_i - \frac{1}{S_i}$ with $u_i$ being the vacuity and $S_i = \sum_{v=1}^{V} e_{iv} + 1$ and $b_i^{inc}$ is the total belief associated with the non-gt tokens.*

The above lemma demonstrates how we assign importance to the two different types of samples. First, the formulation shows that for a higher $\gamma$ we give emphasis to samples with high vacuity, indicating that we attempt to learn from the samples where the model lacks knowledge. Second, the model also focuses on the samples where the incorrect belief is high, meaning that the model focuses on correcting the samples that are incorrect and wrong. As such, the model prevents the samples from being confidently wrong, leading to improved calibration. This phenomenon leads to the following theorem.

**Theorem 3.2** *Let $\mathcal{L}^{EU-Ensemble}(\Theta)$ be the evidential uncertainty-guided ensemble loss and $\mathcal{L}^{ERM}(\Theta)$ being the autoregressive loss then the former is upper-bounded by the latter loss, negative of the entropy, and the normalization constant term. Mathematically, we can write the following*

$$\mathcal{L}^{EU-Ensemble}(\Theta) \geq \sum_{e=1}^{E} [\mathcal{L}_e^{ERM}(\Theta) - \gamma_e \mathcal{H}^e[\hat{p}] + \gamma_e C] \tag{7}$$

*where $E$ being the number of models used in the ensemble, $\mathcal{H}^e$ be the entropy and $C$ is the normalization constant and $\gamma_e$ being the hyperparameter associated with the ensemble model $e$.*

Please refer to the Appendix C for the proof of the Lemma 3.1 and Theorem 3.2.

**Remark.** When we optimize the ensemble loss, the model is striving to make a balance between minimizing the autoregressive loss, and maximizing the entropy. In case of the confident wrong samples, as per the Equation 6, for the higher $\gamma$ value, the model aggressively lowers the confidence score as more emphasis is given in terms of maximizing the entropy. In contrast, for the lower $\gamma$ value, the model focuses mostly toward maximizing the confidence score of the

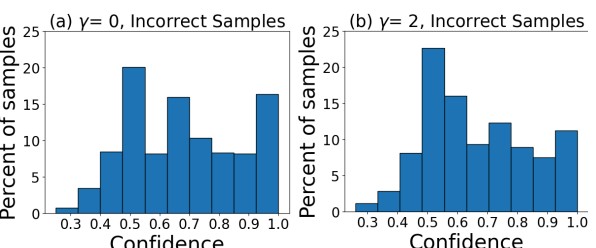

Figure 3: Effect of $\gamma$. As shown for the lower $\gamma$ value i.e., $\gamma = 0$ value the confidences shifted right compared to the one with relatively higher $\gamma$ value i.e., $\gamma = 2$.

samples as the entropy coefficient will be lower. With a high $\gamma$, the model will be generally under-confident, whereas with a low $\gamma$ the model will be overconfident. It is also demonstrated empirically in the Figure 3. As a result, when we take ensemble of the complementary models, they compensate each other resulting into the calibration model.

### 3.4 ADAPTIVE SAMPLING FOR COST REDUCTION

To obtain confidence, it requires $N$ answer samples for the same query. This results in an expensive inference process. We also observe through experiments that the calibration performance improves with the increase in $N$. To address the issue, we propose to adaptively use $N_{adaptive} < N$ for each query such that calibration performance is maintained while reducing the overall cost of the inference. For a question, we iteratively get a sample and update the confidence. In each iteration, we keep track of the change of confidence ($\Delta s$) with the previous iteration. Once the $N_{adaptive}$ reaches $N$ or the

Table 1: Perform comparison on three benchmark datasets and two LVLMs

| Baseline | CommonsenseQA | | | Commonsense Reasoning | | | ScienceQA | | | Compute |
|---|---|---|---|---|---|---|---|---|---|---|
| | ECE | AUROC | Acc. | ECE | AUROC | Acc. | ECE | AUROC | Acc. | |
| **Model: Gemma-4b-it** | | | | | | | | | | |
| Least Probability | 0.262 | 60.294 | 81.980 | 0.500 | 52.430 | 73.540 | 0.285 | 63.880 | 83.447 | $\mathcal{O}(1)$ |
| Product Probability | 0.464 | 62.240 | 81.980 | 0.690 | 54.980 | 73.540 | 0.366 | 62.178 | 83.447 | $\mathcal{O}(1)$ |
| Perplexity | - | 63.060 | 81.980 | - | 54.930 | 73.540 | - | 62.178 | 83.447 | $\mathcal{O}(1)$ |
| Lexical Similarity | - | 61.010 | 82.470 | - | 53.380 | 81.320 | - | 67.480 | 82.100 | $\mathcal{O}(N)$ |
| Semantic Entropy (beam) | 0.108 | 69.220 | 82.470 | 0.117 | 71.510 | 82.470 | 0.091 | 85.690 | 83.560 | $\mathcal{O}(N)$ |
| Multi-output (top-k) | 0.132 | 70.737 | 82.473 | 0.046 | 79.561 | 82.473 | 0.059 | 87.410 | 83.490 | $\mathcal{O}(N)$ |
| Multi-output (Temperature Sampling) | 0.151 | 63.578 | 82.391 | 0.057 | 80.953 | 82.801 | 0.082 | 84.368 | 83.046 | $\mathcal{O}(N)$ |
| Multi-output (nucleus) | 0.146 | 67.884 | 81.572 | 0.031 | 80.965 | 82.228 | 0.063 | 85.414 | 83.565 | $\mathcal{O}(N)$ |
| Vanilla Ensemble | 0.099 | 77.929 | 82.555 | 0.048 | 81.945 | 82.801 | 0.054 | 89.850 | 83.540 | $\mathcal{O}(N)$ |
| DRO Ensemble | **0.051** | **82.445** | 83.047 | **0.017** | **82.568** | 82.801 | **0.040** | **90.375** | 84.485 | $\mathcal{O}(N)$ |
| **Model: LLaVA-OneVision-Qwen2-0.5B** | | | | | | | | | | |
| Least Probability | 0.168 | 65.503 | 63.390 | 0.420 | 52.710 | 58.390 | 0.108 | 75.980 | 75.147 | $\mathcal{O}(1)$ |
| Product Probability | 0.168 | 67.020 | 63.390 | 0.569 | 53.027 | 58.390 | 0.108 | 75.980 | 75.147 | $\mathcal{O}(1)$ |
| Perplexity | - | 67.029 | 63.390 | - | 53.024 | 58.390 | - | 75.223 | 75.147 | $\mathcal{O}(1)$ |
| Lexical Similarity | - | 65.940 | 63.390 | - | 53.350 | 58.720 | - | 68.183 | 76.161 | $\mathcal{O}(N)$ |
| Semantic Entropy (beam) | 0.317 | 65.336 | 63.390 | 0.359 | 61.460 | 58.720 | 0.187 | 68.194 | 76.160 | $\mathcal{O}(N)$ |
| Multi-output (top-k) | 0.195 | 71.307 | 62.817 | 0.117 | 72.107 | 57.330 | 0.103 | 75.810 | 74.794 | $\mathcal{O}(N)$ |
| Multi-output (Temperature Sampling) | 0.241 | 68.006 | 63.718 | 0.190 | 71.541 | 58.313 | 0.121 | 72.610 | 75.925 | $\mathcal{O}(N)$ |
| Multi-output (nucleus) | 0.156 | 71.521 | 63.800 | 0.085 | 72.158 | 56.921 | 0.089 | 76.465 | 75.218 | $\mathcal{O}(N)$ |
| Vanilla Ensemble | 0.170 | 72.881 | 62.735 | 0.065 | 71.165 | 57.821 | 0.078 | 76.757 | 75.595 | $\mathcal{O}(N)$ |
| DRO Ensemble | **0.040** | **73.189** | 63.145 | **0.048** | **72.338** | 57.903 | **0.051** | **78.921** | 74.534 | $\mathcal{O}(N)$ |

$\Delta s \leq \epsilon$, the iteration stops. Reaching the stopping point is when we observe all the samples to have a consistent probability, or the budget $N$ is exhausted. $\epsilon$ is the parameter to control the convergence criteria. The proposed adaptive sampling is agnostic to the multi-output method of obtaining samples.

## 4 EXPERIMENTS

**Datasets, models, and evaluation metrics.** We conduct experiments on three Multiple Choice Questions(MCQ) datasets: ScienceQA Lu et al. (2022), CommonsenseQA Talmor et al. (2019) and Commonsense Reasoning Rajani et al. (2019). Although we consider MCQ datasets, the number of tokens in output varies (ScienceQA: $\approx 200$ tokens, CommonsenseQA: $\approx 1$ tokens, and Commonsense reasoning: $\approx 10$ tokens on average), and answer retrieval is a key part to obtain the most consistent answer. We leverage the Natural Language Inference model (deberta-base-mnli) He et al. (2021) to find whether the obtained answer is semantically consistent with the options. More details about answer retrieval are presented in the Appendix.

We finetune two LVLMs for our experiments: LLaVA-OneVision-Qwen2-0.5B Li et al. (2024) and gemma-3-4b-it Google (2024) with a train split of each of the datasets. For the evaluation of calibration, we leverage Expected Calibration Error (ECE) as a metric. For fault tolerance performance, we leverage AUROC, which measures the area under the curve for a binary classification problem of detecting correct/incorrect test generation using confidence score. We also report the accuracy of each comparison. More details on experiments are presented in the Appendix.

**Comparison baselines.** We compare with both single and multi-output baselines. We include semantic entropy Kuhn et al. (2023a) and lexical similarity Fomicheva et al. (2020) for comparison. We also leverage other decoding strategies to generate $N$ samples (top-k sampling, nucleus sampling Holtzman et al. (2019), and temperature sampling) as baselines. We include vanilla ensemble as a baseline. Similarly, for a single-output method, we include the least probability, product probability Ren et al. (2022), and perplexity as baselines. Details about the least and product probability are included in the Appendix. As perplexity and lexical similarity do not provide probability outputs, we do not report ECE for these baselines.

### 4.1 CALIBRATION AND FAULT TOLERANCE PERFORMANCE

We present the performance of the proposed DRO ensemble and baseline methods in Table 1. Across three different datasets and 2 models of varying range, we observe that the calibration (ECE) and fault tolerance performance (AUROC) of the proposed method are superior in comparison to baselines while maintaining accuracy and inference compute ($\mathcal{O}(N)$). In general, it is seen that multi-output-based methods demonstrate competitive performance. An ensemble of models trained with different seeds (vanilla ensemble baseline) shows improvement over other baselines across multiple settings.

The proposed ensemble further enhances the diversity among ensemble components and outperforms the vanilla ensemble method.

Table 2: Calibration performance comparison of adaptive sampling

| Baseline | $\epsilon$ | CommonsenseQA | | Commonsense Reasoning | |
|---|---|---|---|---|---|
| | | % Samples | ECE | % Samples | ECE |
| Top-k | - | 100.00 | 0.132 | 100.00 | 0.046 |
| Top-k + Adaptive | 0.1 | 42.162 | 0.137 | 50.041 | 0.097 |
| Top-p | - | 100.00 | 0.146 | 100.00 | 0.031 |
| Top-p + Adaptive | 0.1 | 42.129 | 0.150 | 48.010 | 0.089 |
| Vanilla Ensemble | - | 100.00 | 0.099 | 100.00 | 0.048 |
| Vanilla Ensemble + Adaptive | 0.1 | 44.341 | 0.112 | 48.485 | 0.087 |
| DRO Ensemble | - | 100.00 | 0.051 | 100.00 | 0.017 |
| DRO Ensemble + Adaptive | 0.1 | 46.716 | 0.067 | 50.762 | 0.029 |
| DRO Ensemble + Adaptive | 0.01 | 61.556 | 0.066 | 74.758 | 0.024 |

**Adaptive Sampling Performance:** In this section, we present the ECE results of the proposed adaptive function in two datasets using the Gemma3-4b-it model. Along with the proposed DRO Ensemble, adaptive sampling can be leveraged with other multi-output baselines as well. The results for gemma-4b-it are presented in Table 2. The row with $100\%$ samples is the non-adaptive version of each method. For the proposed DRO ensemble, we show the calibration performance with two epsilon values 0.1 and 0.01. When we use $\epsilon = 0.1$, with only $46\%$ of the total samples, we can achieve competitive calibration performance. Similarly, with $\epsilon = 0.01$, with $61\%$ of the total samples, we can further improve calibration performance. The adaptive sampling method can be applied to other sampling and vanilla ensemble methods. For the other baselines, we observe similar competitive calibration performance while reducing the samples. For the Llava model, consistent results are demonstrated, which are presented in Table 10 in the Appendix.

## 4.2 ABLATION STUDY

**Impact of $N$:** In this section, we investigate the effect of varying $N$ (the number of samples per question) on calibration performance for the proposed methods, along with competitive multi-output baselines. The comparison result on the common-sense reasoning dataset by sampling $N = \{24, 21, 18, 12, 9, 6\}$ samples per query is presented in Figure 4. For all methods, as we increase the value of $N$, we have comparable or better calibration performance (decreasing trend of ECE). The results demonstrate an improvement in confidence capability as the value of $N$ increases. Next, we compare performance among

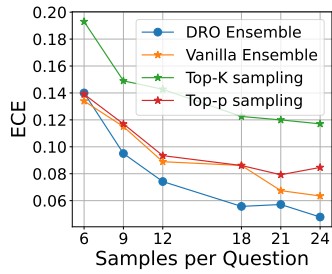

Figure 4: Impact of N

baselines. For all the values of $N$, we achieve performance improvement (ECE) of the proposed ensemble method over the compared baselines. This comparison further demonstrates the effectiveness of the proposed DRO-based ensemble in calibration performance across different values of $N$.

**Impact of $\gamma$:** We study the impact of $\gamma$ on the accuracy and calibration performance of individual models and the proposed DRO ensemble. The study conducted on the CommonsenseQA dataset and Llavaonevision model is presented in Table 3. When we use different values of $\gamma$, we observe that higher values of gamma result in a well-calibrated lower accuracy model, as evidenced by the result of $\gamma = 100$. Increasing the value of $\gamma$ allows the model to focus on more difficult samples, resulting in inaccurate results for easy samples. Our theoretical analysis suggests that the ensemble approach that focuses on a diverse range of difficulty results in more calibrated models. This is also observed from experiments. If we ensemble similar values of $\gamma \in \{1, 2, 5\}$, the calibration performance is still poor.

Table 3: Impact of $\gamma$

| Setting | ECE | Acc. |
|---|---|---|
| $\gamma = 1$ | 0.160 | 63.309 |
| $\gamma = 2$ | 0.165 | 63.554 |
| $\gamma = 5$ | 0.156 | 63.391 |
| $\gamma = 50$ | 0.097 | 61.998 |
| $\gamma = 100$ | 0.044 | 50.942 |
| $\gamma \in \{1, 2, 5\}$ | 0.161 | 63.145 |
| $\gamma \in \{1, 2, 100\}$ | 0.056 | 63.473 |
| $\gamma \in \{1, 50, 100\}$ | 0.043 | 62.817 |
| $\gamma \in \{2, 50, 100\}$ | 0.043 | 62.244 |
| $\gamma \in \{5, 50, 100\}$ | 0.040 | 63.145 |

However, as we include $\gamma = 100$ in the ensemble, the performance significantly improves while maintaining the accuracy. The $\gamma$ values of $\{5, 50, 100\}$ result in the most optimal performance. The inclusion of diverse values of $\gamma$ also shows competitive performance. This study demonstrates the importance of $\gamma$ for the proposed DRO ensemble.

**Calibration Error:** As an ablation study, we leverage the Brier score (Brier, 1950) as a metric to measure calibration error. The Brier score for a generative test dataset can be calculated as, Brier Score $= \sum_{k=1}^{K}(p_k - y_k)^2$. Here, $y_k$ is the ground truth label of whether the sample belongs to class $K$ or not, and $p_k$ is the probability of the sample belonging to class $k$. The comparison of the proposed method is presented along with competitive baselines for the CommonsenseQA dataset in Table 4. Using the Brier score, we observe that the proposed ensemble sampling achieves competitive performance over baselines, consistent with the ECE-based calibration error metric.

Table 4: Comparison of Brier score

| Method | Gemma | LLaVa |
|---|---|---|
| Beam Search | 0.069 | 0.133 |
| Top-p | 0.060 | 0.113 |
| Top-k | 0.062 | 0.108 |
| Vanilla Ensemble | 0.054 | 0.109 |
| DRO Ensemble | 0.049 | 0.102 |

**Impact of individual components:** We conduct an ablation to evaluate the performance gain from individual components. In particular, we compare the calibration performance of a vanilla ensemble, a DRO ensemble without evidential uncertainty, and the proposed method for Commonsense Reasoning with LLaVA-OneVision-Qwen2-0.5B in Table 5. We observe that using a vanilla ensemble of models obtains improved calibration performance of the multi-output method. DRO-based ensemble shows further improvement. DRO guided by evidential uncertainty demonstrates the best calibration performance.

Table 5: Impact of components

| Setting | ECE |
|---|---|
| Multi-output (nucleus) | 0.085 |
| Vanilla Ensemble | 0.065 |
| DRO w/o Uncertainty | 0.057 |
| Proposed Ensemble | 0.048 |

**Impact of $V'$:** In equation 3, we use $V'$ to evaluate the vacuity and belief. In evidential theory, $V'$ refers to the number of classes in the standard classification setting. For token generation, the total number of classes is $V$, which is normally a large value (e.g., 32000 for LLaVA-OneVision-Qwen2-0.5B). As a result, the two uncertainty values become $u_t \approx 1, b_t^{inc} \approx 0$ when $V' = V$, resulting in uniform weights for all the samples. As a practical solution, we used $V' < V$ for the experiments. To be specific, we set $V'$ as the total number of tokens with a positive logit, instead of a constant value for all tokens. In this way, the resulting uncertainties reflect the actual difficulty of learning and result in a calibrated output from the ensemble. To demonstrate the impact of using different values of $V'$, we experiment with OneVision-Qwen2-0.5B model and the Commonsense reasoning dataset. Calibration comparison is presented in Table 6. Using $V' = V(32000)$ demonstrates approximately similar performance to vanilla ensemble. Lowering the value of $V'$ to 100 and 50 gives an improvement in calibration. Setting $V'$ as the number of tokens with positive logits for every token generation demonstrates the superior performance.

Table 6: Impact of V'

| Setting | ECE |
|---|---|
| Vanilla Ensemble | 0.065 |
| $V' = V$ | 0.061 |
| $V' = 100$ | 0.057 |
| $V' = 50$ | 0.052 |
| Ours | 0.048 |

## 4.3 QUALITATIVE STUDY

We present a qualitative analysis of the confidence quantified for an incorrect sample by the baseline and proposed methods. The demonstrated study is a case where even after 25 samples are generated for the same question using a multi-output method from nucleus sampling, the model outputs the same wrong answer of B every 24 times and only 1 different answer. It is illustrated in Figure 5. However, with our method, the model outputs different answers, lowering the confidence score from 0.96 to 0.40. While the prediction from both cases is incorrect, lower confidence from the proposed method improves the calibration. Further, it can be noted that no decoding path to answer the question leads to option A with the multi-option method, while our method obtains answers from diverse paths.

Table 7: Results on 7b and 12b models

| Setting | ECE | Acc. |
|---|---|---|
| **Model: Gemma-3-12b-it** | | |
| Multi-output (nucleus) | 0.047 | 86.077 |
| Vanilla Ensemble | 0.036 | 86.159 |
| Ours | 0.033 | 86.323 |
| **Model: llava-onevision-qwen2-7b-ov** | | |
| Multi-output (nucleus) | 0.042 | 84.029 |
| Vanilla Ensemble | 0.039 | 84.357 |
| Ours | 0.033 | 84.439 |

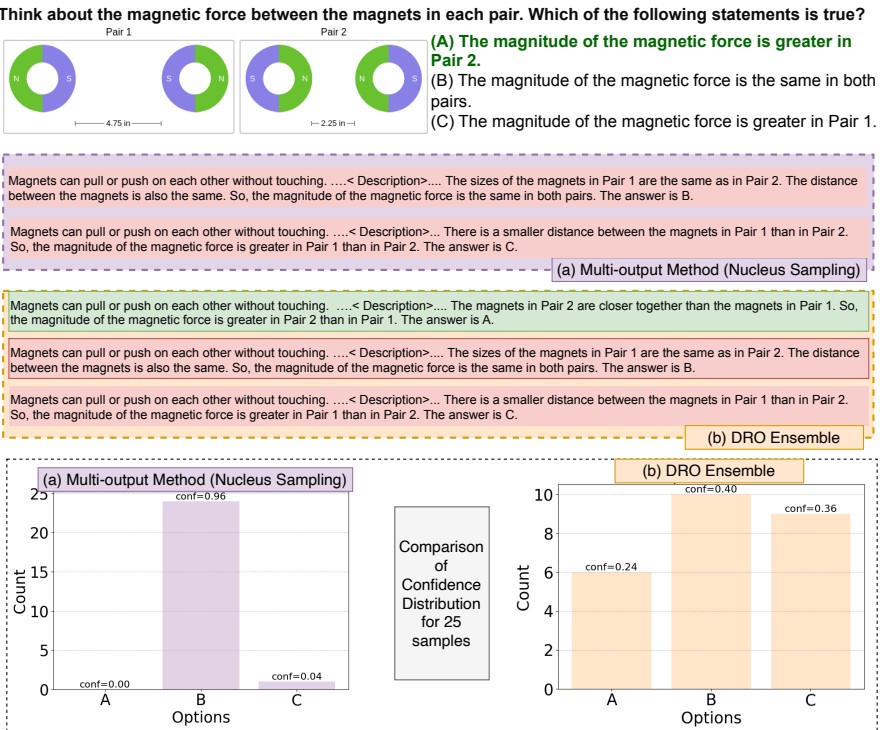

Figure 5: For the question, the prediction from both methods is incorrect. Using (a) Multi-output method (nucleus sampling): the model outputs the wrong answers with high confidence, (b) DRO Ensemble: Diversity components provide diverse answers with low confidence

## 4.4 Additional Results

**Results on 7b and 12b models:** We conduct experiments on Commonsense Reasoning with models: google/gemma-3-12b-it and lmms-lab/llava-onevision-qwen2-7b-ov. The experiment results of our method, along with vanilla ensemble and multi-output nucleus-based sampling baselines, are presented in Table 7. The experiments demonstrate the effectiveness of the proposed method using larger-sized LLMs as well.

Table 8: Results on triviaQA

**Results on open-ended generations:** We conduct an experiment on TriviaQA with gemma-3-4b-it model, an open-ended QA setting. For the generated answers with our method and the baselines, we aggregated them into $K$ unique answers by leveraging the semantic equivalence decision from an NLI model. The results are presented in Table 8. The proposed method demonstrates superior calibration performance over the competitive baselines.

| Setting | ECE | Acc. |
|---|---|---|
| Multi-output (nucleus) | 0.065 | 85.880 |
| Vanilla Ensemble | 0.061 | 86.750 |
| Ours | 0.063 | 87.085 |

## 5 Conclusion

In this work, we study the reliability of large language models from a confidence and calibration perspective. First, we propose a Bayesian grounded, diversity-inducing ensemble guided by the principle of DRO and evidential theory. Next, we provide theoretical guarantees about the proposed method to improve calibration. From the experiments, we demonstrate the effectiveness of the proposed method over baselines. We also perform analysis on the impact of the number of samples $N$, calibration error, individual components of the proposed formulation, and hyperparameters $\gamma$, $V'$ and $\epsilon$. The study of $\gamma$ further demonstrates the diversity mechanism with different combinations of $\gamma$. Finally, to reduce the inference cost while maintaining reasonable calibration performance, we propose an adaptive function. Additional experiments on larger VLMs (7B and 12B models) and open-ended generation setting further demonstrate the effectiveness of the proposed method.

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

# Supplementary Material

# Appendix

**Organization of Appendix.** In this Appendix, we first summarize the major notations used in our paper in Section A. We present Bayesian perspective of multi-output methods in Section B. We provide the detailed mathematical proofs for our theoretical contributions in Section C. Next, we provide the details of the experiment in Section D and additional experiments in Section E. We discuss LLMs usage in Section F. We discuss limitations in Section G and societal impact in Section H. We provide the link to the code in Section I.

## A   SUMMARY OF NOTATIONS

Table 9 summarizes the major notations used in our paper.

Table 9: Symbols with Descriptions

| Symbol Group | Notation | Description |
|:---:|:---:|:---:|
| Dataset | $\{x_{\text{query}}, \mathbf{y}\}$ 
 $x_{\text{image}}$ 
 $x_{\text{text}}$ 
 $y_t$ | Input output pair 
 Image input 
 Text input 
 t-th token |
| Calibration Error | $K$ 
 $\mathcal{Y}_k$ 
 $p_k$ 
 $N$ | Total classes 
 Output set of class $k$ 
 Probability of class $k$ 
 Total number of sampled output |
| Proposed Ensemble | $\mathcal{E}$ 
 $V$ 
 $\boldsymbol{e}_{tv}$ 
 $\boldsymbol{b}_{tv}$ 
 $u_t$ 
 $b_t^{inc}$ 
 $u$ 
 $b^{inc}$ 
 $w_i$ 
 $\gamma$ 
 $\mathcal{L}^{ERM}$ 
 $\mathcal{L}^{robust}$ | Non-negative transformation function 
 Vocabulary size 
 Evidence of $v^{th}$ in $t^{th}$ generation 
 Belief of $v^{th}$ in $t^{th}$ generation 
 Uncertainty of $t^{th}$ generation 
 Incorrect belief of $t^{th}$ generation 
 Uncertainty of generation 
 Incorrect belief of generation 
 Weight of loss in $i^{th}$ training sample 
 Hyperparameter to control $w_i$ 
 Auto-regressive loss 
 Proposed DRO Loss |
| Adaptive Sampling | $N_{adaptive}$ 
 $\epsilon$ | Number of adaptive sampled output 
 Parameter to control convergence |

## B  BAYESIAN PERSPECTIVE OF MULTI-OUTPUT METHODS

Assume a pre-trained LVLM with parameters $\theta$. Given an input query, multiple outputs $\mathcal{Y} = \{\mathbf{y}_n\}_{n=1}^{N}$ are obtained by leveraging probabilistic decoding mechanisms. Aggregating these outputs to obtain a consistent output $\tilde{\mathbf{y}}$ is obtained through marginalization.

$$p(\tilde{\mathbf{y}}_k|x_{\text{query}}) \approx \sum_{\mathbf{y} \in \mathcal{Y}} \frac{|\mathbf{y} \equiv \tilde{\mathbf{y}}_k|}{|\mathcal{Y}|}, \quad \hat{\mathbf{y}} = \arg\max_{k \in K}(p(\tilde{\mathbf{y}}_k|x_{\text{query}})), \quad s = \max_{k \in K}(p(\tilde{\mathbf{y}}_k|x_{\text{query}})) \tag{8}$$

Here, $N$ output answers in answer set $\mathcal{Y}$ can be aggregated to $K$ unique answers $\tilde{\mathcal{Y}} = \{\tilde{\mathbf{y}}_k\}_{k=1}^{K}$. The probability distribution over $k^{th}$ unique label is represented by $p(\tilde{\mathbf{y}}_k|x_{\text{query}})$. It is defined as the fraction of the counts of answers that are semantically equivalent to the $\tilde{\mathbf{y}}_k$ out of all the answers in $\mathcal{Y}$. Semantic equivalence is represented by $\mathbf{y} \equiv \tilde{\mathbf{y}}$.

Performing finetuning of the model with parameter-efficient fine-tuning approaches results in updated parameters $\theta' = \theta + \Delta\theta$. Following the Bayes theorem (where *posterior $\propto$ prior $\times$ likelihood*), using $P(\theta)$ as a prior results in a posterior distribution $p(\theta|D)$ by finetuning with Dataset $D$. A true posterior is intractable and approximated by a distribution $q(\theta)$. Obtaining multiple outputs using model parameters as a random variable $\theta$ can be represented as:

$$p(\tilde{\mathbf{y}}_k|D, x_{\text{query}}) = \int_{\theta} p(\tilde{\mathbf{y}}_k|\theta, D, x_{\text{query}})p(\theta|D)d\theta \approx \sum_{\mathbf{y} \in \mathcal{Y}} \frac{|\mathbf{y} \equiv \tilde{\mathbf{y}}_k|}{|\mathcal{Y}|} \tag{9}$$

Here, $\mathcal{Y}$ is obtained by decoding mechanisms sampling from the updated model $\theta'$. To better approximate the posterior distribution, an ensemble of models finetuned on a pre-trained model could be used. Each ensemble component model parameters $\{\theta^{(m)}\}_{m=1}^{M}$ can be interpreted as $\theta^{(m)} \sim q(\theta)$. For the practical purpose, $M < N$ ensemble components could be combined with decoding-based sampling to obtain $N$ outputs for the query. Our proposed framework guides a loss function to obtain $M$ ensemble components such that $q(\theta)$ is better approximated.

## C  MATHEMATICAL PROOFS

In this section, we provide the mathematical proof for Lemma 3.1 1 and Theorem 3.2.

### C.1  PROOF OF LEMMA 3.1

In this section, we show the that focal loss weight can be decomposed as combination of the vacuity and incorrect below. Focal loss weight is defined as

$$w_i = (1 - \hat{p}_{ik})^{\gamma} \tag{10}$$

From the Subjective logic, we leverage the following relationship.

$$\hat{p}_{ik} = \frac{\hat{\alpha}_{ik}}{S_i} = \frac{\hat{b}_{ik}S_i + 1}{S_i} \tag{11}$$

Using Equation 11, we update the Equation 10 as follows:

$$w_i = \left(1 - \frac{\hat{b}_{ik}S_i + 1}{S_i}\right)^{\gamma} \tag{12}$$

$$= \left(1 - \hat{b}_{ik} - \frac{1}{S_i}\right)^{\gamma} \tag{13}$$

Where $\hat{b}_{ik}$ is the belief associated with the ground truth token $k$ and $S_i = \sum_{j=1}^{V} e_{ij} + 1$. Next, using the standard evidential learning equation:

$$u_i + \sum_{j=1}^{V} b_{ij} = 1 \tag{14}$$

$$u_i + \sum_{j=1, j \neq k}^{V} b_{ij} + \hat{b}_{ik} = 1 \tag{15}$$

$$u_i + \sum_{j=1, j \neq k}^{V} b_{ij} = 1 - \hat{b}_{ik} \tag{16}$$

Now, using the value of $1 - \hat{b}_{ik}$ from Equation 16 to simplify Equation 13, we have:

$$w_i = \left( u_i + \sum_{j=1, j \neq k}^{V} b_{ij} - \frac{1}{S_i} \right)^{\gamma} \tag{17}$$

In Equation 17, the second term refers to the sum of the beliefs associated with the incorrect classes. Denoting $b_i^{inc} = \sum_{j=1, j \neq k}^{V} b_{ij}$, we have the following

$$w_i = \left( u_i + b_i^{inc} - \frac{1}{S_i} \right)^{\gamma} \tag{18}$$

Representing $\hat{u}_i = u_i - \frac{1}{S_i}$ we have the following

$$w_i = (\hat{u}_i + b_i^{inc})^{\gamma} \tag{19}$$

This completes our Lemma 3.1

### C.2    PROOF OF THEOREM 3.2

In the Lemma 3.1 we have shown that focal loss weight being combination of the vacuity and incorrect belief. In this Theorem, we show that our ensemble model loss is upper bounded by the autoregressive loss and entropy. Specifically, following Mukhoti et al. (2020) Appendix, for each model we can write the following:

$$\mathcal{L}^{robust} \geq \mathcal{L}^{ERM} + \gamma \mathcal{H}^e[\hat{p}_y] + \gamma C \tag{20}$$

where $C$ is constant for which above inequality holds true for all ensemble base models. If $a > b$, $c > d$ then we can write $a + b > c + d$. Applying this to the E ensemble models, we can write the following

$$\mathcal{L}^{EU-Ensemble}(\Theta) \geq \sum_{e=1}^{E} [\mathcal{L}_e^{ERM}(\Theta) - \gamma_e \mathcal{H}^e[\hat{p}] + \gamma_e C] \tag{21}$$

This completes the Theorem 3.2

## D    DETAILS OF EXPERIMENTS

**Datasets:**    We conduct experiments on three datasets. ScienceQA is a multimodal dataset across 26 topics of science where answer contains multiple-choice option, along with explanations ($\approx 200$ tokens on average). CommonsenseQA is a text-only dataset built on a large commonsense knowledge graph where answer contains multiple choice options ($\approx 1$ tokens on average). Commonsense reasoning is a text-only dataset where the answer contains multiple choice options, along with a natural language explanation ($\approx 10$ tokens on average).

**Models and Implementation Details:** We use two models, gemma-3-4b-it and LLaVA-OneVision-Qwen2-0.5B-ov. We use pretrained models from Huggingface with repository names: google/gemma-3-4b-it and lmms-lab/llava-onevision-qwen2-0.5b-ov. For experiments, we use the device with a GPU: NVIDIA RTX A6000 and memory: 48 GB. We perform Lora fine-tuning of the models using the training split as provided in the original splits, and present the results for the test split. We follow the fine-tuning instructions from repositories Li. (2024); won Lee. (2024). For each ensemble component, we use nucleus sampling with values: $top\_p = 0.9$ and temperature $= 0.7$. We provide the detailed code to reproduce the experiments in section I.

**Answer Retrieval:** To check whether the answer corresponds to one of the options in the multiple choice question, we use the text containing the obtained answer and the multiple choice option as an input to a natural language inference model. Given $x_{query}$ with options, and the generated answer $\mathbf{y}$, we use the following text:

$$\text{TEXT} = <x_{query}> + <\mathbf{y}> + [\text{SEP}] + <x_{query}> + <\text{Multiple Choice Option}> \tag{22}$$

NLI model returns the logits for 3 classes. If the obtained answer $\mathbf{y}$ entails the multiple choice option, the logit is highest for $3^{rd}$ class. Based on the logit, we can retrieve the answer. To accurately retrieve the answer, it is recommended to use both the character and actual answer in the <Multiple Choice Option>.

**Baselines:** For top-p sampling, we use the value of $top\_p = 0.9$. For beam sampling, we use $num\_beams = 5$. For temperature sampling, we use the value of temperature as $0.7$. For top-k sampling, we use the value of $top\_k = 50$. For all multi-output baselines except temperature sampling, we use the value of temperature as $1.0$. For the least probability, we obtain confidence as the minimum of the probability of all generated tokens, as $\min_{t=1}^{T} p(y_t|y_{<t}, x)$. For the product probability, we multiply the probability of all generated tokens as $\prod_{t=1}^{T} p(y_t|y_{<t}, x)$.

**Expected Calibration Error:** Given $N$ from decoding methods, we have an answer set $\mathcal{Y}$. We place each answer into the respective multiple-choice option, aggregating them to $K$ unique answers $\tilde{\mathcal{Y}} = \{\tilde{\mathbf{y}}_k\}_{k=1}^{K}$. The probability distribution over $k^{th}$ unique label is represented by $p(\tilde{\mathbf{y}}_k|x_{query})$. Confidence score and predicted label are given by:

$$p(\tilde{\mathbf{y}}_k|x_{query}) \approx \sum_{\mathbf{y} \in \mathcal{Y}} \frac{|\mathbf{y} \equiv \tilde{\mathbf{y}}_k|}{|\mathcal{Y}|}, \quad \hat{\mathbf{y}} = \arg\max_{k \in K}(p(\tilde{\mathbf{y}}_k|x_{query})), \quad s = \max_{k \in K}(p(\tilde{\mathbf{y}}_k|x_{query})) \tag{23}$$

For a test dataset $\{x^i, y^i, \hat{\mathbf{y}}^i\}_{i=1}^{N'}$, where $x^i$ is $i^{th}$ dataset input, $y^i$ is the corresponding ground truth label, and $\hat{\mathbf{y}}^i$ is the prediction from the model. First, we bin the dataset into $B$ equally spaced confidence intervals Naeini et al. (2014); Nixon et al. (2019). For each bin $B_b$ accuracy$(B_b)$ denotes the accuracy of the dataset belonging to the bin $B_b$, and confidence$(B_b)$ denotes the average confidence of the dataset belonging to the bin $B_m$. The expected calibration error is calculated as the mean of the difference between accuracy and confidence across all the bins.

$$ECE = \sum_{b=1}^{B} \frac{|B_b|}{N'} |\text{accuracy}(B_b) - \text{confidence}(B_b)| \tag{24}$$

For ECE calculation, we use the bin size of 10.

# E  ADDITIONAL EXPERIMENTS

## E.1  ADAPTIVE SAMPLING

In the main paper, we demonstrate the effectiveness of adaptive sampling for gemma-4b-it model. In this section, we study the performance of adaptive sampling for the proposed DRO ensemble,

along with other competitive baselines for lava-onevision-qwen2-0.5b-ov model. The results are presented in Table 10. For the proposed ensemble, when $\epsilon = 0.1$ is used, with only 50% of the samples, the calibration performance is competitive compared to the non-adaptive setting. When we use $epsilon = 0.01$, the calibration performance further improves. Other competitive baselines, in comparison to the proposed DRO ensemble, perform poorly. When compared with its own non-adaptive version, these methods also demonstrate reasonable calibration performance upon leveraging adaptive sampling.

Table 10: Calibration performance comparison of adaptive sampling for lava-onevision-qwen2-0.5b-ov model

| Baseline | $\epsilon$ | CommonsenseQA | | Commonsense Reasoning | |
|---|---|---|---|---|---|
| | | % Samples | ECE | % Samples | ECE |
| Top-k | - | 100.00 | 0.195 | 100.00 | 0.117 |
| Top-k + Adaptive | 0.1 | 46.47 | 0.209 | 50.172 | 0.140 |
| Top-p | - | 100.00 | 0.156 | 100.00 | 0.085 |
| Top-p + Adaptive | 0.1 | 47.486 | 0.174 | 50.925 | 0.104 |
| Vanilla Ensemble | - | 100.00 | 0.170 | 100.00 | 0.065 |
| Vanilla Ensemble + Adaptive | 0.1 | 47.715 | 0.172 | 50.631 | 0.102 |
| DRO Ensemble | - | 100.00 | 0.040 | 100.00 | 0.048 |
| DRO Ensemble + Adaptive | 0.1 | 50.106 | 0.077 | 50.713 | 0.091 |
| DRO Ensemble + Adaptive | 0.01 | 87.109 | 0.067 | 87.224 | 0.068 |

## E.2 RESULTS ON POPE DATASET

In this section, we present the comparison of calibration performance in POPE dataset Li et al. (2023c) for the proposed ensemble sampling and multi-output method using nucleus sampling. For fine-tuning, we use "popular" split from the dataset to train the Lora component for gemma3-4b-it model. For testing, we use "random" split. The comparison is presented in Figure 7b for $N \in \{24, 21, 18\}$. Using the proposed ensemble sampling, the calibration performance improves over the multi-output method. This study further confirms the superior calibration performance of the proposed method over the multi-output method.

## E.3 COMPARISON WITH POST-HOC CALIBRATION METHODS

We performed a post-hoc temperature scaling to improve calibration performance on Commonsense reasoning with the Google/Gemma-3-4b-it model. For the held-out dataset, we searched for the temperature parameter that obtains the optimal Negative Log Likelihood loss of the output generation. For a generated output $\mathbf{y} = [y_1, y_2, ...y_T]$, the dimension of the logits is $T \times V$. In case of the model considered, the size of the vocabulary is $V = 262, 208$. The results of the experiment are presented in the Table 11.

Table 11: Comparison with post-hoc calibration

| Setting | w/o Temperature Scaling | with Temperature Scaling |
|---|---|---|
| Single Output | 0.690 | 0.679 |
| Multi-Output (nucleus) | 0.031 | 0.366 |

We observe a marginal improvement with temperature scaling. Our proposed method significantly outperforms the performance with and w/o temperature scaling. Similarly, we also performed temperature scaling for the multi-output-based baseline. With $N$ generations of output, we aggregate them to $K$ multiple-choice answers. We treat the logit for each option as the number of answers semantically equivalent to the option, $\sum_{\mathbf{y} \in \mathcal{Y}} |\mathbf{y} \equiv \tilde{\mathbf{y}}_k|$. The temperature scaling with the multi-output method significantly degrades the performance. In the case of MCQ datasets, although we have $K$ options, the actual answer varies from one sample to another. Hence, searching for an optimum temperature for all the data samples does not lead to optimum calibration performance for the whole dataset.

### E.4 RESULTS ON LIMITED DATA SETTING

We experimented on a limited data setting by using only $5\%$ and $8\%$ of training samples of the Commonsense Reasoning dataset with the LLaVA-OneVision-Qwen2-0.5B model. By comparing the calibration performance of the proposed method with vanilla ensemble and multi-output based method, we observe the effectiveness of the proposed method in a limited data setting as well. The results are presented in Table 12.

Table 12: Results on limited data

| Setting | 5 % | 8 % |
|---|---|---|
| Multi-output (nucleus) | 0.115 | 0.099 |
| Vanilla Ensemble | 0.106 | 0.092 |
| Ours | 0.100 | 0.083 |

### E.5 HANDLING LONG-TAIL DATA DISTRIBUTION

In our setting, the weighting term $w_i$ in Equation 4 enables the model to adaptively focus more on difficult samples while assigning less emphasis to easier ones. For challenging samples, $w_i$ becomes large either due to high vacuity or because more belief mass is placed on incorrect classes, compared to easier cases. Consequently, during backpropagation, the model prioritizes reducing the loss associated with these difficult samples.

Importantly, DRO does not encourage overfitting; rather, it helps prevent it through the implicit regularization induced by the entropy term. As established in Theorem 3.2, our loss decomposes into the cross-entropy minus the entropy, where the negative entropy term discourages the model from producing incorrect, overconfident predictions.

To empirically verify the effectiveness of the proposed method over baselines for handling long-tail data distribution, we conducted experimen on the ScienceQA dataset with the Gemma3-4b-it model. We fine-tune the model with the minitrain split of the dataset. While the original minitrain split contains 115 samples from the "magnet" category, we include only 4 samples from the same category, making it a tail class. We present the performance comparison between the proposed method and baselines in the Table 13.

Table 13: Results on long tailed data

| Setting | ECE | Acc. |
|---|---|---|
| Multi-output (nucleus) | 0.116 | 76.415 |
| Vanilla Ensemble | 0.106 | 74.764 |
| Ours | 0.059 | 77.358 |

The results contain both the calibration and accuracy performance for the minitest split. We observe a clear performance gain of the proposed method over baselines. Further, we compare performance over only the tail class, where we leverage a total of 110 samples from "magnet" category. The results are presented in the Table 14.

Table 14: Results on tail class "magnet"

| Setting | ECE | Acc. |
|---|---|---|
| Multi-output (nucleus) | 0.333 | 37.273 |
| Vanilla Ensemble | 0.305 | 31.818 |
| Ours | 0.164 | 44.545 |

The accuracy performance of all methods is lower compared to the whole minitest split. However, this is expected, as we leverage very few samples in training. When we compare the calibration performance among the methods, our method shows significant improvement over the baselines. These results provide important empirical evidence to demonstrate the robustness of the proposed method for long-tail data.

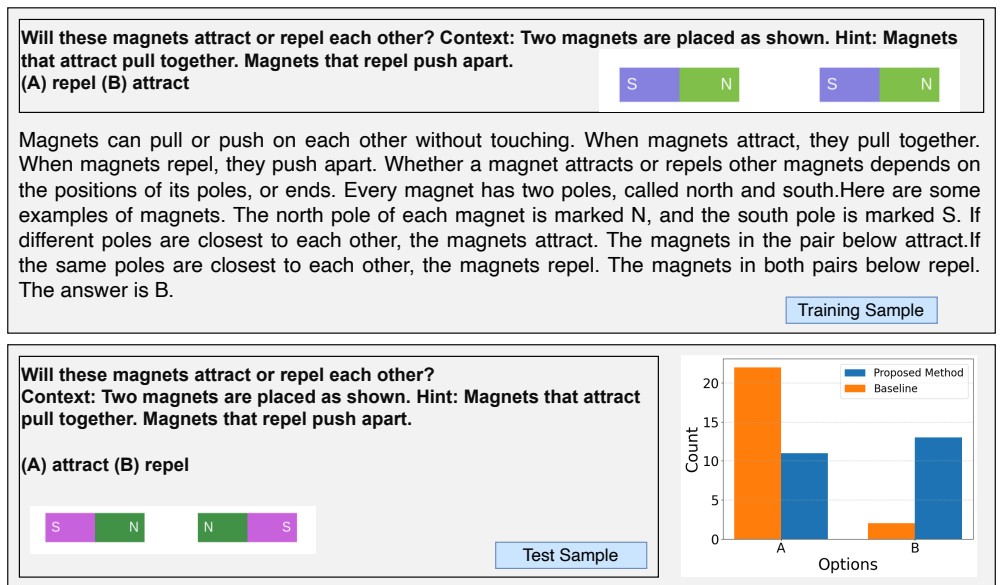

Figure 6: Qualitative analysis of tail class result in long tailed data study.

Finally, we present a qualitative analysis in the Figure 6. We have a training sample where the question is about whether magnets "repel" or "attract" each other. In this specific sample, we have the case of "attract" (north and south poles facing each other). In testing, we present a similar question, but with north poles facing each other (i.e., the correct answer being "repel"). The baseline method (multi-output with nucleus sampling) outputs answer of "attract" 22 out of 24 times, making a confidently wrong prediction. In contrast, proposed method outputs answer of "repel" 13 out of 24 times. When only few samples of "magnet" category is shown in training, the baseline method suffers from overfitting and results in overconfident predictions. The proposed method demonstrates superior calibration over the baseline.

## E.6 ABLATION STUDIES

**Impact of $N$:** In the main paper, we study the impact of $N$ for the commonsense reasoning dataset in lava-onevision-qwen2-0.5b-ov model. In this section, we conduct the study for the commonsenseQA dataset. Along with the proposed DRO method, we also include competitive baselines in the study. The results are presented in Figure 7a. We observe that as the value of $N$ increases, the calibration performance improves. The observed pattern is consistent for all the studied methods.

**Impact of calibration error:** In the main paper, we present the results of CommonsenseQA by leveraging the Brier Score as a metric to measure calibration error. In this section, we present results of DRO Ensemble along with competitive multi-output based methods on Commonsense Reasoning and ScienceQA datasets in the Table 15. Leveraging the Brier score further validates the superior calibration performance of the proposed method.

**Impact of $\epsilon$:** The hyperparameter $\epsilon$ is set to stop the iteration in adaptive sampling. A lower value of $\epsilon$ results in early stopping of iteration and generates a lower percentage of samples. The comparison of different values of $\epsilon = \{0.5, 0.1, 0.05, 0.01, 0.005, 0.001\}$, along with calibration performance and generated percentage of samples, is presented in the Table 16. As we lower the value of $\epsilon$, we generate more samples, resulting in a higher calibration performance.

**Proposed Ensemble Formulation:** In this section, we conduct the ablation study on the proposed ensemble. Each ensemble component is trained using the loss function guided by uncertainty. First, we study the impact of different uncertainties: $u$ and $b^{inc}$ in Equation 1. We re-formulate the weight

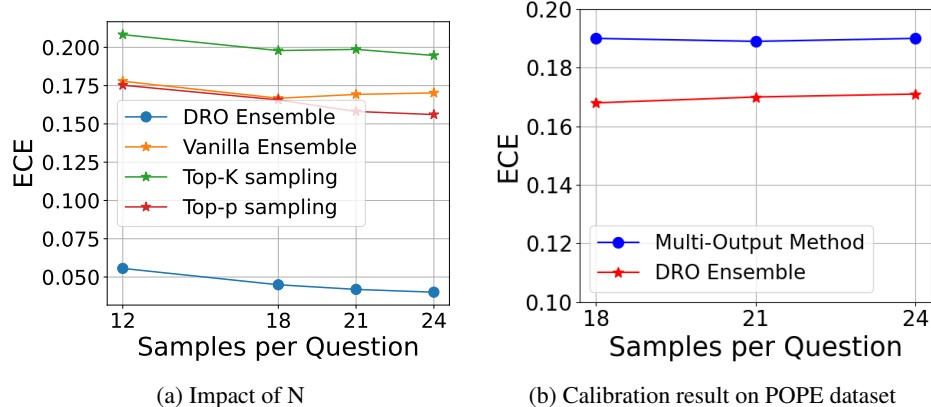

(a) Impact of N          (b) Calibration result on POPE dataset

Figure 7: Additional result: (a) Impact of N for commonsenseQA dataset (b) ECE comparison of multi-output method (nucleus sampling) and proposed DRO ensemble for pope dataset

Table 15: Impact of calibration error for Commonsense Reasoning and ScienceQA datasets

|  | **Commonsense Reasnoning** | | **ScienceQA** | |
| **Method** | **Gemma** | **LLaVa** | **Gemma** | **LLaVa** |
| --- | --- | --- | --- | --- |
| Beam Search | 0.059 | 0.150 | 0.046 | 0.084 |
| Top-p | 0.050 | 0.114 | 0.045 | 0.073 |
| Top-k | 0.053 | 0.111 | 0.046 | 0.070 |
| Vanilla Ensemble | 0.049 | 0.112 | 0.044 | 0.070 |
| DRO Ensemble | 0.049 | 0.111 | 0.041 | 0.069 |

Table 16: Impact of $\epsilon$

| $\epsilon$ | % of samples | ECE |
| --- | --- | --- |
| 0.5 | 31.581 | 0.118 |
| 0.1 | 50.713 | 0.091 |
| 0.05 | 68.436 | 0.064 |
| 0.01 | 87.224 | 0.068 |
| 0.005 | 92.580 | 0.053 |
| 0.001 | 93.120 | 0.053 |
| – | 100.00 | 0.040 |

as $w_i = (a_1 u + (1 - a_1)b^{inc})^\gamma$. We vary the value of $a_1 \in \{0.1, 0.3, 0.5, 0.7, 0.9\}$ for $\gamma = 2$. For each setting, we sample $N$ samples per query from two ensemble components. The first component is trained with $w_j = 1$ and the second component is trained with $w_i = (a_1 u + (1 - a_1)b^{inc})^2$. For a multi-output method, all the $N$ answers are sampled from the model trained with $w_i = 1$. Next, we study the impact of using a loss function guided by probability. For product probability-based focal loss, we use the value of $w_i = (1 - \prod_{t=1}^{T} p(y_t|y_{<t}, x))^2$, and for min probability-based focal loss, we use the value of $w_i = (1 - \min_{t=1}^{T} p(y_t|y_{<t}, x))^2$.

We experiment on the minitest split of ScienceQA. The results are presented in Figure 9. All the settings exhibit better calibration performance than the multi-output baseline. Among the values of $a_1$, the settings with the lower value of $a_1$ demonstrate better ECE values. Specifically, for $N < 18$, $a_1 = 0.1$ has the best ECE, and for $N \geq 18$, $a_1 = 0.7$ has the best ECE. Lower values of $a_1$ are the settings where we put more weight on incorrect beliefs than on uncertainty. Incorrect beliefs refer to the model's beliefs about the incorrect labels. Penalizing those beliefs is essential for the model to be accurate. It should be noted that too much penalty on incorrect belief does not always result in best calibration performance(as demonstrated by best performance $a_1 = 0.7, N \geq 18$).

Think about the magnetic force between the magnets in each pair. Which of the following statements is true?

(A) The magnetic force is stronger in Pair 1.
**(B) The magnetic force is stronger in Pair 2.**
(C) The strength of the magnetic force is the same in both pairs.

Pair 1 | Pair 2

⊢ 2 in ⊣ | ⊢ 1 in ⊣

Magnets can pull or push on each other without touching. ....< Description>... But the distance between the magnets in Pair 1 and in Pair 2 is the same. So, the strength of the magnetic force is the same in both pairs. The answer is C.

(a) Multi-output Method (Nucleus Sampling)

Magnets can pull or push on each other without touching. ....< Description>.... The magnets in Pair 1 are closer together than the magnets in Pair 2. So, the magnetic force is stronger in Pair 1 than in Pair 2. The answer is A.

Magnets can pull or push on each other without touching. ....< Description>.... The magnets in Pair 2 are closer together than the magnets in Pair 1. So, the magnetic force is stronger in Pair 2 than in Pair 1. The answer is B.

Magnets can pull or push on each other without touching. ....< Description>... But the distance between the magnets in Pair 1 and in Pair 2 is the same. So, the strength of the magnetic force is the same in both pairs. The answer is C.

(b) DRO Ensemble

Figure 8: Qualitative analysis: For the question, the prediction from both methods is incorrect. Using (a) Multi-Output Method (Nucleus Sampling): the model answers the wrong answers with high confidence, (b) DRO Ensemble: Diversity components provide diverse answers with low confidence

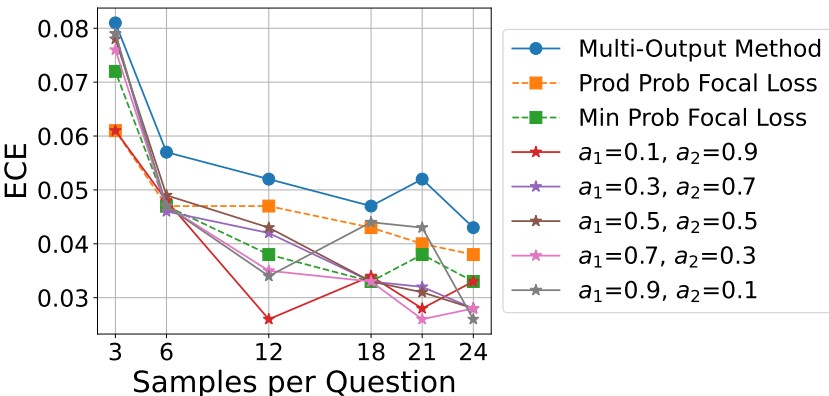

Figure 9: Ablation of the proposed ensemble: Impact of putting weights $a_1$ and $a_2$ on uncertainty $u$ and incorrect belief $b^{inc}$ on $w_j$ of the loss function, and impact of using a different focal loss function

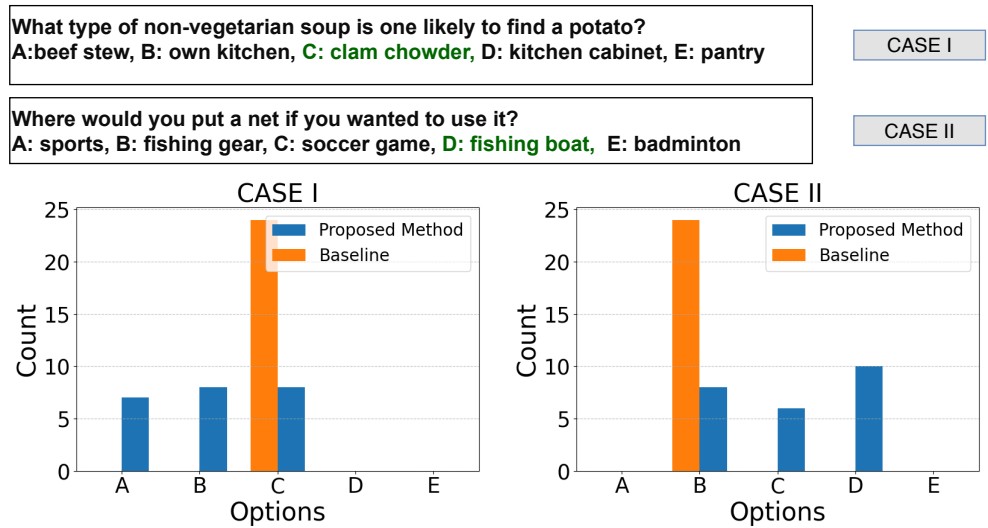

Figure 10: Qualitative analysis of the error case of the proposed method. Case I: The Proposed method is wrong, and the baseline is correct. Case II: The Proposed method is correct, and the baseline is wrong.

Among min-prob and prod-prob based focal loss, we have superior performance of the min-prob based focal loss. Minimum probability among tokens helps the model to focus on the most uncertain tokens. This study further justifies the design of the proposed loss function for guiding the model to focus on the key token while fine-tuning. When compared with other settings, min-probability-based focal loss performs comparably to the setting that puts equal weight on uncertainty and incorrect belief. High incorrect beliefs and high uncertainty are two contributing factors for a low probability value. Loss function based on probability score does not allow fine-grained control on the source of low probability in the training ensemble component.

### E.7 QUALITATIVE ANALYSIS

In this section, we present a case of qualitative analysis: (i) the advantage of using the proposed ensemble over multi-output sampling in Figure 8. We observe that for the question asked, even after 24 samples, the multi-output method with nucleus sampling confidently provides the wrong output without leading to the diverse paths of A and B. In contrast, the proposed method outputs other, more diverse answers, lowering the confidence of the provided output from 1.0 to 0.79. In this way, the proposed method improves the calibration of generative content.

### E.8 ANALYSIS OF LOWER ACCURACY

We observe that we achieve lower accuracy in some cases in Table 1. We conducted an analysis of the method with highest accuracy baseline(baseline: beam sampling: 58.720) vs (our method: 57.903) for Commonsense reasoning and LLaVA-OneVision-Qwen2-0.5B. While the baseline achieves marginally higher accuracy, the probability assigned is mostly on the higher end, resulting in untrustworthy confidence when the answer is wrong. On contrary, our method does not concentrate probability on the higher end, resulting in lower confidence when the answer is wrong. This makes the proposed method more trustworthy. This is also demonstrated by ECE and AUROC (baseline: 0.359, 61.460) and (our method: 0.048, 72.338) in Table 1. We show two qualitative analyses of different cases. First, we present the test sample, where our method is wrong, but the baseline is right. In the second case, we present the test sample, where our method is correct, and the baseline is wrong. It is presented in Figure 10. In case I, option C is correct. Baseline answers option C: 24 times. Our method answers option A: 7 times, B: 8 times and C: 8 times. Our method is wrong, but

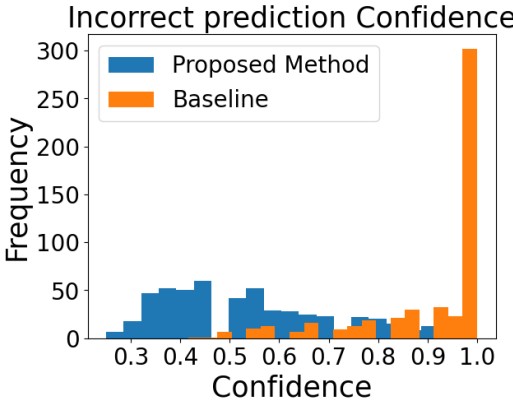

Figure 11: Confidence distribution of the proposed method and the baseline with the highest accuracy on incorrect predictions

the confidence is low (0.3). In case II, option D is correct. Baseline answers option B: 24 times. Our method answers option B: 8 times, C: 6 times and D: 10 times. Baseline is confidently wrong.

We plot a histogram of confidence of incorrect predictions from the proposed method and the baseline in Figure 11. For the baseline, the confidence of wrong predictions is concentrated towards the higher end. The confidence distribution further reveals the trustworthiness of the proposed method over the most accurate baseline.

## F  USAGE OF LLMS

LLM tools have been utilized only to polish writing, not for generating ideas, methods, or results. Tools were used to help debug Python errors while creating the plots for the paper.

## G  LIMITATIONS

In this work, we propose a novel diversity-inducing ensemble with theoretical guarantees. We demonstrate the effectiveness of the proposed method in multiple-choice datasets. We consider settings with multiple output tokens as output; the answer belongs to one of the multiple-choice options. We leverage the entailment method to retrieve the answer given an output. We also experimented with an open-ended QA (TriviaQA), demonstrating the effectiveness of the proposed method. For open-ended QA, we placed answers into the same group if they were semantically equivalent to each other. We will perform a more systematic evaluation of open-ended settings as future work.

## H  SOCIETAL IMPACT

Large language models are becoming increasingly popular in a lot of applications. Although they can perform many tasks, the content generated is not always correct and trustworthy. The concern becomes more pronounced in critical applications (medical, education). The model can quantify a confidence or uncertainty score along with the generated content. The confidence can be used to decide whether we should trust the generated content. The research in this work is performed to ensure that the generated content has a calibrated confidence, such that quantified confidence correlates with correctness of the output.

## I  SOURCE CODE

The source code is available at this link.

## J  COMPUTATIONAL COST ANALYSIS

The computational analysis in Table 1 is for inference. We include the computational analysis to separate the inference cost of single-output: $\mathcal{O}(1)$ and multi-output methods: $\mathcal{O}(N)$. In our experiments, we generate a constant number of samples $N$ for all multi-output methods, resulting in the same inference cost for our method and vanilla ensemble as the multi-output baselines.

Training ensemble components requires an extra cost for the proposed method. We reduce the cost of training ensemble components by leveraging $M < N$ models for the proposed method. To ensure that $M$ components cover the diversity of models to approximate the posterior, we propose a DRO-based ensemble guided by evidential uncertainty. Further, we leverage a parameter-efficient fine-tuning method to fine-tune the pretrained model to reduce the training cost. Fine-tuning of the ensemble components with downstream task is a one-time expense and the ensemble members can be run in parallel across multiple GPUs for computational efficiency.

