# OpenReview forum: "Bayesian-Informed Diverse Sampling for Calibration of Fine-Tuned Foundation Models with Evidential Ensembles"
_ICLR.cc/2026/Conference — Submitted to ICLR 2026_

### Official Review · Reviewer_CzTL · 2025-10-26

**Soundness:** 3
**Presentation:** 3
**Contribution:** 2
**Rating:** 4
**Confidence:** 3

**Summary:**

This paper addresses the problem of unreliable and overconfident outputs from fine-tuned foundation models, especially under domain shift and when downstream fine-tuning data are limited or noisy. The authors observe that commonly used single-output or single-score uncertainty estimators (token-level probabilities, model self-reports, or embedding probes) are fragile. Multi-output approaches, which marginalize across decoding paths or measuring disagreement across multiple generations, can help, but still remain insufficient when the posterior over model parameters $p(\theta)$ becomes flat or multimodal after fine-tuning with limited/noisy data.

The paper proposes a Bayesian-informed, diversity-driven ensembling method that (1) deliberately creates an ensemble whose members emphasize different subsets of training samples (simple vs. difficult) using Distributionally Robust Optimization (DRO), and (2) computes sample-wise difficulty and ensemble uncertainty using an Evidence Theory formulation applied at the token level. The combination yields an ensemble that is both diverse, which benefits from fine-tuning with different focuses, and calibrated for detecting unreliable outputs.

The contributions of this article include: (1) It introduces the Evidence Theory to explicitly separate and quantify two failure modes, i.e., lack of knowledge (uncertainty mass $u$ ) and confusion among labels (incorrect belief $b^{inc}$ ), rather than treating all uncertainty as a single scalar. (2) It proposes a simple, scalable method to combine the Evidence Theory with DRO for creating diverse, calibration-aware ensembles by training multiple fine-tuned models with different difficulty-weighting hyperparameters $\gamma$  and combining their outputs with evidence-theoretic uncertainty scoring. This paper also uses adaptive sampling to reduce computation. (3) The approach targets practical failure modes of fine-tuned foundation models and demonstrates improved detection of incorrect outputs and overall calibration compared to single-output, multi-output, and naive ensemble baselines.

Overall, the paper offers a theoretically guaranteed and practically applicable framework to make fine-tuned large foundation models more robust and better calibrated by combining evidence-based token-level uncertainty decomposition with ensemble diversity driven by DRO.

**Strengths:**

(1) The authors integrate principles from Bayesian modeling, DRO, and evidential theory to design an ensemble approach that targets the limitations of pure decoding-based or randomly initialized ensembles. The theoretical development ties the motivation to Bayesian marginalization and calibration guarantees.

(2) The decomposition of sample difficulty using fine-grained evidential uncertainty (vacuity and incorrect belief) is insightful and novel in the context of generative models.

(3) The methodology is explicit, with the loss and weighting strategy clearly defined, and the impact of the diversity parameter ($\gamma$ ) on ensemble behavior is both theoretically analyzed and empirically studied.

(4) The experimental setup is robust. It includes multiple multiple choice questions benchmark datasets and two different LVLMs, and adopts several uncertainty quantification and calibration metrics. The paper also provides a detailed ablation study.

**Weaknesses:**

(1) The novelty is limited. This paper combines the prior DRO-based ensemble methods (Sapkota et al., 2023) with the Evidence Theory (Dempster-Shafer Theory). Although the combination is insightful, its core idea is not entirely new. The authors should clarify to what extent the evidence-based weight formulation is new, and provide evidence (either theoretical or empirical) that the proposed method offers advantages that are not achievable by existing approaches.

(2) Some directly related works are neither discussed, nor compared in the experiments, e.g., Multi-class calibration [1] and gradient diversity in ensembles [2].

(3) The choice of some key hyperparameters, such as $V'$ in Equation 3 and $\epsilon$ in the adaptive sampling, is neither well justified nor thoroughly explored.

(4) To demonstrate broader generalization, the method should be evaluated on other generation tasks, such as open‑ended QA (especially multi‑hop reasoning), math QA, code generation, and other natural language generation benchmarks. These evaluations would strengthen claims about applicability and robustness.

References
[1] Zhao, Shengjia, et al. "Calibrating predictions to decisions: A novel approach to multi-class calibration." Advances in Neural Information Processing Systems 34 (2021): 22313-22324.
[2] Trinh, Trung Q., et al. "Input gradient diversity for neural network ensembles." CoRR (2023).

**Questions:**

(1) Could the authors elaborate on the motivation for using DRO and the Dempster–Shafer Evidence Theory, and the point of putting these ideas together? A more detailed motivation would help readers understand the design choices behind the method.

(2) Please provide a detailed statement of the default value of the vocabulary cutoffs ($V'$) used in Equation 3, and the reason for that choice. Could you further provide a quantitative sensitivity analysis for it? In particular, how does this choice affect calibration error and accuracy?

(3) The proposed method does not achieve the highest accuracy (Table 1, 3). Can the authors provide a more detailed failure mode analysis?

(4) The proof of Lemma 3.1 appeals to a relationship from Subjective Logic but omits several derivation steps. Please expand the proof and show the key intermediate steps and assumptions.

(5) The discussion of computational overhead in Table 1 seems insufficient. Ensemble-based methods require each of the $M$ components to generate $N$ responses, implying an $O(MN)$ generation cost.

(6) In the experiments, the authors use a Natural Language Inference model (deberta-base-mnli) to capture semantical consistency between the answer and the options, and the answer aggregation is natural given the options. How to handle the semantic equivalence and answer aggregation in more practical settings where options are not provided, such as multi-hop reasoning or open-ended natural language generation tasks?

Typo:
Line 277: "and and"

---

> ### Author Response · Authors · 2025-11-21
> **Rebuttal by Authors**
>
> Thank you for taking the time to review our paper. We appreciate your constructive comments. Our responses are summarized below.
>
> **Q1: Clarify to what extent the evidence-based weight formulation is new, and provide evidence (either theoretical or empirical) that the proposed method offers advantages that are not achievable by existing approaches.**
>
> Uncertainty quantification in generative language modeling remains an open challenging research area. Our work makes three novel contributions to advance this important field.
>
> - As our first contribution, we introduce a DRO-guided evidential ensemble framework that accurately approximates the posterior distribution of ensemble members, enabling more reliable and fine-grained uncertainty estimation. To the best of our knowledge, this is the first evidentially backed ensemble approach designed specifically to enhance uncertainty quantification.
>
> - Second, beyond providing better uncertainty estimates, we present a theoretically grounded DRO ensemble formulation that provably improves the calibration accuracy of generative models. This is a non-trivial advance, as VLMs especially when hallucinating often remain highly confident due to strong linguistic priors. For example, even when an image clearly contains a black banana, a model may still assert that the banana is yellow because its language prior overwhelms the visual evidence. Such prior-driven responses lead the model to be confidently wrong. Our framework reduces this overconfidence in hallucination scenarios through DRO-based ensembling, as demonstrated by the substantial reduction in ECE reported in Table 1.
>
> - Third, we establish a solid theoretical foundation by showing that our robust loss function forms an upper bound on the sum of the ERM objective and negative entropy (which guarantees improved calibration). We believe this is a meaningful and non-trivial extension in the VLM setting, as it integrates the distinct principles of evidential learning and focal-style robustness to deliver uniquely improved calibration.
>
> **Q2: References not discussed**
>
> Thank you for suggesting these important related work. We provide a discussion of the related papers and highlight their differences from our method as follows:
>
> - [1] proposes a post-hoc recalibration algorithm for multi-class classifiers. It introduces decision calibration to ensure that a model's predicted probabilities are indistinguishable from true probabilities for decision-making tasks. Their algorithm uses a pre-trained, fixed predictor and iteratively adjusts its outputs. In each step, it identifies the worst violation of decision calibration (i.e., a partition of the prediction space where predicted and true probabilities diverge most) and applies a corrective update to the predictions for that partition. This process is repeated until the model satisfies decision calibration, without altering the original model weights.
>
> - [2] performs calibration at the gradient level during training. It uses an explicit repulsion force between ensemble members. This force is not applied in the weight space but in the input-gradient space. The core idea is to push the gradients of each model's output with respect to the input data away from each other, forcing each network to learn different explanatory features for the same data point. This encourages functional diversity by ensuring each model relies on different input patterns, effective for improving robustness and calibration under covariate shifts.
>
> Overall, [1] focuses on post-hoc recalibration without changing original model weights, and [2] focuses on gradient-level manipulation to boost diversity for calibration. In contrast, our method introduces an uncertainty-aware fine-tuning methodology to train a diverse ensemble, with unique loss functions guided by Distributionally Robust Optimization and evidential theory. This loss function uses a parameter to weight training samples based on their difficulty (measured by evidential uncertainty). The ensemble components are forced to specialize in different difficulty levels, thereby creating a diverse ensemble that better approximates the model's complex posterior distribution and improves calibration. We added these references in the Relation Work (Page 3, Section 2)  of the revised manuscript.
>
>  [1] Zhao, Shengjia, et al. "Calibrating predictions to decisions: A novel approach to multi-class calibration." Advances in Neural Information Processing Systems 34 (2021): 22313-22324.
>
>  [2] Trinh, Trung Q., et al. "Input gradient diversity for neural network ensembles." CoRR (2023).
>
> **Q3: Experiments on open-ended generation settings**
>
> Thank you for this valuable comment. We provide experiment results on an open-ended QA dataset (TriviaQA) in Q1 of general response.

---

> ### Author Response · Authors · 2025-11-21
> **Rebuttal by Authors**
>
> **Q4: Impact of hyperparameter $V'$**
>
> Equation 3 in the paper uses $V'$ for the quantification of vacuity and belief. In evidential theory, $V'$ refers to the number of classes while performing classification. In the case of token generation, the total number of classes is $V$, which is normally a large value (e.g., $32000$ for LLaVA-OneVision-Qwen2-0.5B). As a result, the two uncertainty values become $u_t \approx 1, b_t^{inc} \approx 0$ when $V' = V$, resulting in uniform weights for all the samples. As a practical solution, we used $V' <V$ for the experiments. To be specific, we set $V'$ as the total number of tokens with a positive logit, instead of a constant value for all tokens. In this way, the resulting uncertainties reflect the actual difficulty of learning and result in a calibrated output from the ensemble.
>
> To demonstrate the impact of using different values of $V'$, we experiment with OneVision-Qwen2-0.5B model and the Commonsense reasoning dataset. Calibration comparison is presented in the following table. Using $V' = V (32000)$ demonstrates approximately similar performance to vanilla ensemble. Lowering the value of $V'$ to $100$ and $50$ gives an improvement in calibration. Setting $V'$ as the number of tokens with positive logits for every token generation demonstrates the superior performance.
>
> |        Setting            |  ECE   |
> |---------------------------|--------|
> |  Vanilla Ensemble         | 0.065  |
> |  V' = V                   | 0.061  |
> |  V' = 100                 | 0.057  |
> |  V' = 50                  | 0.052  |
> |  Ours                     | **0.048**  |
>
> We have included the discussion about the impact of $V'$ in the revised manuscript (Page 9, Section 4.2, Table 6).
>
> **Q5: Impact of hyperparameter $\epsilon$**
>
> The hyperparameter $\epsilon$ is set to stop the iteration in adaptive sampling. A lower value of $\epsilon$ results in early stopping of iteration and generates a lower percentage of samples. The comparison of different values of $\epsilon = \{ 0.5, 0.1, 0.05, 0.01, 0.005, 0.001 \}$, along with calibration performance and generated percentage of samples, is presented in the following table. As we lower the value of $\epsilon$, we generate more samples, resulting in a higher calibration performance.
>
> |    ε    |  % of samples   |  ECE  |
> |---------|-----------------|-------|
> |  0.5    |     31.581      | 0.118 |
> |  0.1    |     50.713      | 0.091 |
> |  0.05   |     68.436      | 0.064 |
> |  0.01   |     87.224      | 0.068 |
> |  0.005  |     92.580      | 0.053 |
> |  0.001  |     93.120      | 0.053 |
> |   -     |     100.00      | 0.040 |
>
> We included this discussion in the Appendix of the revised manuscript (Page 21, Section E.6, Table 16).
>
>
> **Q6: Lower Accuracy for some cases**
>
> Thank you for pointing out the lower accuracy for some cases. We conducted an analysis of the method with the highest accuracy baseline (baseline: beam sampling: $58.720$) vs (our method: $57.903$). While the baseline achieves marginally higher accuracy, the probability assigned is mostly on the higher end, resulting in untrustworthy confidence when an answer is wrong. On contrary, our method does not concentrate probability on the higher end, resulting in lower confidence when the answer is wrong. This makes the proposed method more trustworthy. This is also demonstrated by ECE and AUROC (baseline: 0.359, 61.460) and (our method: 0.048, 72.338) in Table 1.
>
> We show two qualitative analysis of different cases. First, we present a test sample, where our method is wrong, but the baseline is right. In the second case, we present a test sample, where our method is correct, and baseline is wrong.
>
> - **CASE I:**
>
> >Question: What type of non-vegetarian soup is one likely to find a potato?
>
> >A:beef stew, B: own kitchen, C: clam chowder, D: kitchen cabinet, E: pantry
>
> >In case I, option C is correct. Baseline answers option C: $24$ times. Our method answers option A: $7$ times, B: $8$ times and C: $8$ times. Our method is wrong, but the confidence is low ($0.3$), which indicates good calibration.
>
> - **CASE II:**
>
> > Question: Where would you put a net if you wanted to use it?
>
> > A: sports, B: fishing gear, C: soccer game, D: fishing boat, E: badminton
>
> > In case II, option D is correct. Baseline answers option B: $24$ times. Our method answers option  B: $8$ times, C: $6$ times and D: $10$ times. Baseline is confidently wrong, indicating poor calibration.
>
> We plot a histogram of confidence of incorrect predictions from proposed method and the baseline in Figure 11 of the revised paper. For the baseline, the confidence of wrong predictions is concentrated towards the higher end. The confidence distribution further reveals the trustworthiness of the proposed method over the most accurate baseline.
>
> ... Continued in next comment ...

---

> ### Author Response · Authors · 2025-11-21
> **Rebuttal by Authors**
>
> In Table 3, using different values of $\gamma$ leads to different accuracies. A lower value of $\gamma$ leads to model focusing on the most representative samples. As a result, it performs better in terms of accuracy due to test samples being similar to train sample distribution. As we increase the value of $\gamma$, the model focuses on the most difficult training samples. When we ensemble models focusing on different levels of difficulties, we achieve calibration predictions with comparable accuracy performance.
>
> We included discussion of lower accuracy cases in the Appendix of the revised manuscript (Page 24, Section E.8, Figures 10 and 11).
>
> **Q7: Lemma 3.1 proof**
>
> Thanks for the reviewer's concerns on detailed steps and the assumptions associated with it. First, we would like to emphasize that our Lemma 3.1 is strictly based on the evidential learning and the definition of the belief, and uncertainty are taken directly from the evidential learning framework. Therefore, we have not made any external assumptions deriving this. In terms of the detailed steps, we have revised the paper with the intermediate steps involved in the proof in Section C.1 of the Appendix.
>
> **Q8: Computational complexity analysis**
>
> Thanks for your comment about the complexity cost. We include discussion of computational cost for training and inference in Q2 of the general response.
>
>
> **Q9: Handling semantic equivalence and answer aggregation in more practical settings**
>
> Thank you for the question. For the MCQ dataset, we aggregated answers into $K$ options using an NLI model. For more practical settings (open-ended QA, multi-hop reasoning datasets), the aggregation can be handled using a clustering algorithm. In the experiment of Trivia QA we conducted for Q3, we put two answers in the same cluster group only if they are semantically equivalent to each other.
>
>
> **Q10: Typo**
>
> Thank you for pointing out the typo. We have fixed it in the revised manuscript.

---

### Official Review · Reviewer_AoME · 2025-10-26

**Soundness:** 3
**Presentation:** 3
**Contribution:** 3
**Rating:** 6
**Confidence:** 3

**Summary:**

The paper presents a novel approach to improve the calibration and uncertainty quantification of fine-tuned foundation models in critical domains. It proposes a Bayesian-informed, diversity-inducing ensemble method guided by Distributionally Robust Optimization and evidential theories. The core challenge addressed is the miscalibration and overfitting that often occurs during the fine-tuning of these models, especially when domain shifts and limited downstream data exacerbate the complexity of the model’s posterior distribution.
Empirical results across benchmark datasets and large visual language models demonstrate the effectiveness of the approach in comparison to existing methods, with superior performance in both calibration and fault tolerance. The paper also introduces an adaptive sampling method to reduce the inference cost without sacrificing calibration quality.

**Strengths:**

1. The authors introduce a DRO-based loss function that enhances the diversity among ensemble components, allowing for more accurate uncertainty estimation.
2. The paper provides solid theoretical guarantees for the proposed method's calibration improvements, backed by extensive empirical validation across multiple benchmark datasets and LVLMs.
3. The ability to accurately quantify uncertainty and improve the calibration of large foundation models is crucial for their deployment in critical, high-stakes domains.

**Weaknesses:**

1. please fully dissect the contributions of individual components (e.g., DRO loss, evidential uncertainty) to the overall performance.
2. fine-tuning with limited data is a common real-world challenge, and the paper does not explore this
3. it does not consider long-tail data distributions or rare error cases that are crucial in practice.
4. how the method would perform if γ is misconfigured.

**Questions:**

1. Could the DRO loss function potentially cause overfitting or instability during training, especially in tasks with limited data?
2. DRO ensemble enhances output diversity. How do we ensure that the diversity between the models is not just increasing computational complexity without significantly improving performance?

---

> ### Author Response · Authors · 2025-11-21
> **Rebuttal by Authors**
>
> Thank you for taking the time to review our paper. We appreciate your constructive comments. Our responses are summarized below.
>
>
> **Q1: Contribution of individual components**
>
> We conducted an ablation study of the proposed method to evaluate the performance gain from individual components. In particular, we compare the calibration performance of a vanilla ensemble, DRO without evidential uncertainty, and the proposed method for Commonsense Reasoning with LLaVA-OneVision-Qwen2-0.5B in the following table. We observe that using a vanilla ensemble of models in generation obtains improved calibration performance of the multi-output method. DRO-based ensemble shows further effectiveness. DRO guided by evidential uncertainty demonstrates the best calibration performance.
>
> We added the ablation in the revised manuscript (Page 9, Section 4.2, Table 5).
>
>
> |        Setting                            |  ECE  |
> |-------------------------------------------|-------|
> |  Multi-output (nucleus)                   | 0.085 |
> |  Vanilla Ensemble                         | 0.065 |
> |  DRO Ensemble w/o Evidential Uncertainty  | 0.057 |
> |  Ours                                     | **0.048** |
>
>
> **Q2: Limited data setting**
>
> We experimented on a limited data setting by using only 5% and 8% of training samples of the Commonsense Reasoning dataset with the LLaVA-OneVision-Qwen2-0.5B model. By comparing the calibration performance of the proposed method with vanilla ensemble and multi-output based method, we observe the effectiveness of the proposed method in a limited data setting as well.
>
> We added the experiment results in the Appendix of the revised manuscript (Page 20, Section E.4, Table 12).
>
> |        Setting            |  5%   |  8%   |
> |---------------------------|-------|-------|
> |  Multi-output (nucleus)   | 0.115 | 0.099 |
> |  Vanilla Ensemble         | 0.106 | 0.092 |
> |  Ours                     | **0.100** | **0.083** |
>
>
> **Q3: Handling long-tail data distribution**
>
> In our setting, the weighting term $w_i$ in Eq.~4 enables the model to adaptively focus more on difficult samples while assigning less emphasis to easier ones. For challenging samples, $w_i$ becomes large either due to high vacuity or because more belief mass is placed on incorrect classes, compared to easier cases. Consequently, during backpropagation the model prioritizes reducing the loss associated with these difficult samples.
>
> Importantly, DRO does not encourage overfitting; rather, it helps prevent it through the implicit regularization induced by the entropy term. As established in Theorem~3.2, our loss decomposes into the cross-entropy minus the entropy, where the negative entropy term discourages the model from producing incorrect, overconfident predictions.
>
> To empirically verify the effectiveness of the proposed method over baselines for handling long-tail data distribution, we conducted experiment on the ScienceQA dataset with the Gemma3-4b-it model. We fine-tune the model with the minitrain split of the dataset. While the original minitrain split contains $115$ samples from the "magnet" category, we include only $4$ samples from the same category, making it a tail class.
> We present the performance comparison between the proposed method and baselines in the following table.
>
> |        Method            |  ECE   |   Acc. |
> |---------------------------|-------|---------|
> |  Multi-output (nucleus)   | 0.116 |  76.415 |
> |  Vanilla Ensemble         | 0.106 |  74.764 |
> |  Ours                     | **0.059** |  **77.358** |
>
> The results contain both the calibration and accuracy performance for the minitest split. We observe a clear performance gain of the proposed method over baselines. Further, we compare performance over only the tail class, where we leverage a total of $110$ samples from "magnet" category. The results are presented in the following table.
>
> |        Method            |  ECE   |   Acc. |
> |---------------------------|--------|--------|
> |  Multi-output (nucleus)   | 0.333  | 37.273 |
> |  Vanilla Ensemble         | 0.305  | 31.818 |
> |  Ours                     | **0.164**  | **44.545** |
>
> The accuracy performance of all methods is lower compared to the whole minitest split. However, this is expected, as we leverage very few samples in training. When we compare the calibration performance among the methods, our method shows significant improvement over the baselines. These results provide important empirical evidence to demonstrate the robustness of the proposed method for long-tail data.
>
> ... continued in the next comment ...

---

> > ### Author Response · Authors · 2025-11-23
> > **Rebuttal by Authors**
> >
> > Finally, we present a qualitative analysis in Figure 6 of the revised paper. We have a training sample where the question is about whether magnets "repel" or "attract" each other. In this specific sample, we have the case of "attract" (north and south poles facing each other). In testing, we present a similar question, but with north poles facing each other (i.e., the correct answer being "repel"). The baseline method (multi-output with nucleus sampling) outputs answer of "attract" $22$ out of $24$ times, making a confidently wrong prediction. In contrast, proposed method outputs answer of "repel" $13$ out of $24$ times. When only few samples of "magnet" category is shown in training, the baseline method suffers from overfitting and results in overconfident predictions. The proposed method demonstrates superior calibration over the baseline.
> >
> > We added the experiment results in the Appendix of the revised manuscript (Page 20, Section E.5, Table 13, 14 and Figure 6).
> >
> > **Q4: Impact of $\gamma$**
> >
> > We present the impact study of $\gamma$ in Table 3.  Misconfiguration of $\gamma$ may impact the calibration performance as evidenced by the performance of using only lower values of gamma, i.e., $ \gamma \in \{ 1, 2, 5 \}$.
> >
> > **Q5: Diversity from DRO ensemble**
> >
> > By assigning distinct weights to different training samples, each base learner in a DRO ensemble tend to specialize on different parts of the data distribution (e.g., easy, medium, and difficult samples). When combined, these complementary learners yield significantly improved calibration. Our experimental results in Table 1 further support this finding: the ECE improves substantially (e.g., from 0.17 to 0.04 in the LLaVA-OneVision-Qwen2-0.5B model) without sacrificing task performance.
> >
> > Regarding computational cost, we assume the reviewer is primarily concerned about inference-time overhead due to ensembling. Training-time cost is less of an issue, as the ensemble members can be run in parallel across multiple GPUs and this is a one-time expense. Regarding inference related computational cost, we generate a constant number of samples $N$ for all multi-output methods, resulting in the same inference cost as baselines.
> >
> > Please refer to the discussion of computational cost in Q2 of the general section for more details.

---

### Official Review · Reviewer_Yz1U · 2025-11-03

**Soundness:** 2
**Presentation:** 2
**Contribution:** 2
**Rating:** 4
**Confidence:** 3

**Summary:**

This paper proposes a DRO-evidential ensemble method to improve calibration of fine-tuned foundation models. The key idea is to train M ensemble components with different γ values to focus on varying sample difficulties, where difficulty is measured by evidential uncertainties (vacuity u and incorrect belief b^inc). Experiments on three MCQ datasets show improvements in ECE and AUROC over multi-output baselines.

**Strengths:**

Well-motivated problem: Calibration after fine-tuning is important for deployment safety

Solid empirical results: Consistent ECE improvements across datasets (e.g., 0.099→0.051 on CommonsenseQA)

Adaptive sampling: Practical contribution reducing inference cost by ~50% while maintaining performance

Theoretical attempt: Provides analysis connecting the loss to cross-entropy and entropy regularization

**Weaknesses:**

Limited experimental scope: The paper only evaluates on multiple-choice questions with small models (0.5B, 4B). This severely limits the claims: Real-world LLMs are 7B-70B+ with open-ended generation; MCQ with NLI-based answer retrieval is far from practical generation tasks. Authors acknowledge this ("leave open-ended QA as future work") but it undermines the paper's contribution

Weak theoretical justification: Theorem 3.2 only provides an upper bound, not a guarantee of improved calibration. The choice of w_i = (u + b^inc)^γ appears ad-hoc - why sum? why not product or other combinations? The connection between Bayesian posterior approximation (Section B) and the actual method is tenuous

Missing key comparisons: No comparison with post-hoc calibration methods (temperature scaling, Platt scaling) which are computationally cheaper, or with conformal prediction approaches. The claimed O(N) complexity ignores the M× training cost.

**Questions:**

See above weaknesses.

---

> ### Author Response · Authors · 2025-11-21
> **Rebuttal by Authors**
>
> Thank you for taking the time to review our paper. We appreciate your constructive comments. Our responses are summarized below.
>
>
> **Q1: Experiments on small models only**
>
> Thank you for the comment. In the paper, we conduct experiments with two models of sizes $0.5$B and $4$B, respectively. Following reviewer's comment about the size of real-world LLMs being in the range of $7$B - $70$B, we conducted experiments with google/gemma-3-12b-it and lmms-lab/llava-onevision-qwen2-7b-ov models and commonsense reasoning dataset. The experiment results of our method, along with vanilla ensemble and multi-output nucleus-based sampling baselines, are presented in the following table.  The experiments demonstrate the effectiveness of the proposed method using larger-sized LLMs as suggested by the reviewer.
>
>
> |        Setting       | gemma-12b ECE   |  gemma-12b   Acc. |    llava-7b  ECE      |   llava-7b  Acc.  |
> |----------------------|-------|---------|---------------|-----------|
> |Multi-output (nucleus)| 0.047 | 86.077  |       0.042   |    84.029 |
> |Vanilla Ensemble      | 0.036 | 86.159  |       0.039   |    84.357 |
> |Ours                  | **0.033** | **86.323**  |       **0.033**   |    **84.439** |
>
> We added the experiment results to the revised manuscript (Page 10, Section 4.4, Table 7).
>
> **Q2: Evaluation on open-ended generation settings**
>
> We provide experiment results on an open-ended QA dataset (TriviaQA) in the general section.
>
> **Q3: Weak theoretical justification: Theorem 3.2 only provides an upper bound, not a guarantee of improved calibration.**
>
> Thank you for raising this point. In Theorem 3.2 of the paper, the ensemble loss $\mathcal{L}^\texttt{EU-Ensemble}(\Theta)$  is lower bounded by the right-hand side (RHS) of Equation (7), $\sum_{e=1}^E [\mathcal{L}_e^{ERM}(\Theta)-\gamma_e \mathcal{H}^e[\hat{p}]+\gamma_eC]$. Minimizing the ensemble loss therefore pushes down this RHS, which consists of the ERM losses and a negative-entropy term. For misclassified examples, if we assume the ERM loss cannot be further reduced, the only way to decrease the RHS is to increase the entropy $\mathcal{H}^e[\hat{p}]$. This discourages the ensemble from making highly confident wrong predictions, leading to improved calibration.
>
> **Q4: Formulation of weight $w_i$**
>
> We respectfully disagree with the reviewer’s comment that “the choice of $w_i$ is ad hoc.” The specific form of $w_i$ arises directly from the evidential learning framework and is firmly grounded in the evidential learning theorem [1]. We kindly refer the reviewers to the proof of Lemma 3.1 in Appendix C, which shows why $w_i$ must take the form of a sum of vacuity and incorrect belief up to a constant rather than a product of the two.
>
> **Q5: Bayesian posterior approximation**
>
> The discussion of the Bayesian perspective for fine-tuning with dataset $D$  is guided by Bayes theorem: $\textit{posterior}$ $\propto$ $\textit{prior}$ $\times$ $\textit{likelihood}$, where $p(\theta)$ acts as a prior, resulting in a posterior of $p(\theta |D)$. True posterior is approximated with an approximate posterior distribution $q(\theta)$. Each ensemble component $\theta^{(m)}$ is interpreted to sample from approximate posterior distribution as $\theta^{(m)} \sim q(\theta)$, resulting in the following:
>
> $p(y| D, x)=\int_{\theta} p(y|\theta,D, x) p(\theta|D)\text{d} \theta  = E_{q(\theta)} [p(y|\theta,D, x)] \approx \frac{1}{M} \sum_{m=1}^M p(y|\theta^{(m)}, x) $
>
> In our case, $\theta^{(m)}$ is guided by the principle of Distributionally Robust Optimization and evidential uncertainty. We aggregate output generations $\mathcal{Y}$ into $K$ unique output answers, where $\boldsymbol{y} \in \mathcal{Y},  \boldsymbol{y} \sim p(y| D, x)$ and approximate the output probability distribution, given by the following:
>
>    $p(y_k | D, x_{\text{query}})$ $ \approx \sum_{ \textbf{y} \in \mathcal{Y}} \frac{| \textbf{y} \equiv \tilde{\textbf{y}}_k |}{|\mathcal{Y}|}$
>
> The aggregation is performed to capture the semantic difference in the output distribution of a language model.
>
> **References**:
>
> [1] Sensoy et al. "Evidential Deep Learning to Quantify Classification Uncertainty". NIPS 2018

---

> ### Author Response · Authors · 2025-11-21
> **Rebuttal by Authors**
>
> **Q6: Comparison with post-hoc calibration methods**
>
> As suggested by the reviewer, we performed a post-hoc temperature scaling  to improve calibration performance on Commonsense reasoning with the Google/Gemma-3-4b-it model.  For the held-out dataset, we searched for the temperature parameter that obtains the optimal Negative Log Likelihood loss of the output generation. For a generated output $\textbf{y} = [y_1, y_2, ... y_T]$, the dimension of the logits is $T \times V$. In case of the model considered, the size of the vocabulary is $V= 262,208$.  The results of the experiment are presented in the table below.
>
>
>
> |        Setting        |w/o Temperature Scaling| with Temperature Scaling |
> |-----------------------|-----------------------|--------------------------|
> | Single Output         |       0.690           |      0.679               |
> | Multi-Output (nuclues)|      **0.031**           |      0.366               |
>
>
> We observe a marginal improvement with temperature scaling. Our proposed method significantly outperforms the performance with and w/o temperature scaling. Similarly, we also performed temperature scaling for the multi-output-based baseline. With $N$ generations of output, we aggregate them to $K$ multiple-choice answers. We treat the logit for each option as the number of answers semantically equivalent to the option. The temperature scaling with the multi-output method significantly degrades the performance. In the case of MCQ datasets, although we have $K$ options, the actual answer varies from one sample to another.  Hence, searching for an optimum temperature for all the data samples does not lead to optimum calibration performance for the whole dataset.
>
> We added the experiment results in the Appendix of the revised manuscript (Page 19, Section E.3, Table 11).
>
>
> **Q7: Computational complexity analysis**
>
> Thank you for your comment about the training complexity cost. Please refer to Q2 of general section for the discussion about the complexity analysis of training and inference.

---

### Author Response · Authors · 2025-11-21
**Rebuttal by Authors**

**General Response and Summary of Changes in the Revised Paper**

We thank all the reviewers for their time to review our paper. In this general response, we address some common questions raised by reviewers. Specific concerns are addressed in the corresponding rebuttal.

**Q1: Evaluation on open-ended generation settings**

Following the comments from reviewers Yz1U and CzTL, we conducted an experiment on TriviaQA with google/gemma-3-4b-it model, an open-ended QA setting. For the generated answers with our method and the baselines, we aggregated them into $K$ unique answers, as defined by Eq. (5) in the paper. The results are presented in the following table. The proposed method demonstrates superior calibration performance over the competitive baselines.


|        Setting          |  ECE  |     Accuracy  |
|-------------------------|-------|---------------|
|  Multi-output (nucleus) | 0.096 |       85.880  |
|  Vanilla Ensemble       | 0.070 |       86.750  |
|  Ours                   | **0.063** |       **87.085**  |

We added the experiment results to the revised manuscript (Page 10, Section 4.4, Table 8).

**Q2: Computational complexity analysis**

Thanks reviewers Yz1U and CzTL for your comments about the complexity cost. We would like to clarify that the computational analysis in Table 1 is for inference. We include the computational analysis to separate the inference cost of single-output: $\mathcal{O}(1)$ and multi-output methods: $\mathcal{O}(N)$. In our experiments, we generate a constant number of samples $N$ for all multi-output methods, resulting in the same inference cost for our method and vanilla ensemble as the multi-output baselines.

Training ensemble components require an extra cost for the proposed method. We reduce the cost of training ensemble components by leveraging $M < N$ models for the proposed method. To ensure that $M$ components cover the diversity of models to approximate the posterior, we propose a DRO-based ensemble guided by evidential uncertainty. Further, we leverage a parameter-efficient fine-tuning method to fine-tune the pretrained model to reduce the training cost. Fine-tuning of the ensemble components with downstream task is a one-time expense and the ensemble members can be run in parallel across multiple GPUs for computational efficiency.

We added a section of computational complexity analysis in the Appendix of the revised manuscript (Page 26, Appendix J).

***


Following the concerns of reviewers, we provided the rebuttal of the work. The summary of major changes made in the revised paper includes the following:

- **Experiments with practical settings for model size and dataset**: We presented the comparison of the proposed method and competitive baselines on larger models (7b and 12b models) in Table 7, and open-ended QA setting in Table 8 of the revised paper.


- **Contribution of individual components:** We added an ablation to study the contribution of the DRO ensemble and the evidential uncertainty for calibration performance in Table 5 of the revised paper.

- **Impact of hyperparameters $V'$ and $\epsilon$:** We included the study of hyperparameter $V'$ in Table 6 and $\epsilon$ in Table 16 of the revised paper.

- **Comparison with the post-hoc calibration method**: We added experimental results of the post-hoc calibration method (temperature scaling) in Table 11 of the revised paper.


- **Study of fine-tuning with limited data and long-tail data**: We conducted the study of calibration behavior of methods when fine-tuned with limited data and long-tail data, and the results are discussed in Appendices E.4 and E.5, respectively, of the revised paper.

- **Analysis of lower accuracy**: We conducted an analysis of lower accuracy cases for the proposed method over the most accurate baseline. The discussion is presented in section E.8 of the revised paper.

- **Detailed proof of Theorem 3.1:** We included the details of proof for theorem 3.1 in Section C.1 of the revised paper.

- **Discussion of references**: We included the discussion of additional references mentioned by the reviewer in the related work of the revised paper.

- **Computational cost analysis**: We present a separate Appendix J in the revised paper to discuss the cost of training and inference for the proposed method.

All the changes are highlighted in blue for easy identification by the reviewers.

---

### Author Response · Authors · 2025-12-03
**Summary of rebuttal**

We thank all reviewers for their time and thoughtful evaluation of our paper. In this summary note, we first highlight several major misunderstandings and missing pieces of information that may have caused an unfair evaluation of our work. We then summarize the key changes incorporated into the paper during the rebuttal phase that further strengthen our contributions.

$\textbf{Major misunderstandings / missing key information from the paper}$

$\textbf{Theoretical justification:}$ Reviewer $\textit{Yz1U}$ expressed concerns that our theoretical justification is weak, citing a lack of formal guarantees, an ad-hoc design of the weight function, and no clear connection to Bayesian posterior approximation. This is a clear misunderstanding. In the rebuttal, we clarified that the proposed loss is in fact **lower-bounded** in a way that yields a formal guarantee of improved calibration. We further showed that the weight function is theoretically grounded and established an explicit connection between our formulation and Bayesian posterior approximation.

$\textbf{Novelty of the method:}$ Reviewer $\textit{CzTL}$ questioned the novelty of the proposed approach. In the rebuttal, we clarified the novelty of our formulation by highlighting three novel components: (i) a DRO-guided evidential ensemble framework to approximate the posterior distribution and enhance uncertainty quantification, (ii) a theoretically grounded DRO ensemble formulation that improves calibration performance, and (iii) accompanying theoretical guarantees for the proposed method.

$\textbf{Choice of parameters }$ $V'$ $\textbf{ and }$ $\epsilon$: Reviewer $\textit{CzTL}$ noted that the paper did not specify how $\epsilon$ and $V'$ were chosen. However, we have already provided an analysis on the impact of $\epsilon$ in Table 2, which has been missed by the reviewer.  In the rebuttal, we provided an additional empirical study of $V'$, together with insights into the design choices and their impact on performance, and incorporated this discussion into the revised manuscript.

---

$\textbf{Important extensions via the rebuttal}$

$\textbf{Additional experiments:}$ We extended our empirical evaluation to $\textbf{open-ended generation settings}$, $\textbf{larger models}$ (7B–70B), and practical finetuning scenarios with $\textbf{limited}$ and $\textbf{long-tailed data}$. Across all these settings, we compared our method against strong baselines and observed that it remains robust and effective. We also added an ablation study that quantifies the $\textbf{individual contributions}$ of DRO and evidential uncertainty to the overall calibration performance.

$\textbf{Comparisons with additional baselines:}$ We incorporated experiments with a representative post-hoc calibration method, temperature scaling, and found that it performs notably worse than our method and other competitive baselines, especially in more challenging regimes.

$\textbf{Computational cost:}$ We clarified the meaning of the $\mathcal{O}(N)$ notation in the main results table and provided a more detailed breakdown of computational cost at $\textbf{training}$ and $\textbf{inference}$ time. We also explained how the proposed design controls the overhead and keeps it comparable to standard training and inference pipelines.

$\textbf{Additional related work and discussion of key results:}$ We added discussion of two additional $\textbf{references}$ and clarified how they relate to and differ from our approach. We also analyzed cases where our method shows $\textbf{lower accuracy}$ than the most accurate baseline, providing quantitative and qualitative insights into this behavior and its relationship to improved calibration.

---
Overall, the reviewers have acknowledged that the problem addressed in this paper is well-motivated and that the proposed methodology benefits from strong theoretical guarantees and extensive empirical validation. By incorporating additional experiments on key practical settings, expanding comparisons with further baselines, and refining the discussion, we have further strengthened the paper during the rebuttal. In addition, the enriched related work section, guided by the reviewers’ suggestions, more clearly situates our contributions and underscores the novelty of the proposed approach.

---

### Meta-Review · Area_Chair_WW86 · 2026-01-08

**Summary:**

The paper proposes a new diversity-inducing ensemble approach based on Distributionally Robust Optimization (DRO) and evidential theories. The approach is similar to training a standard ensemble, but for each ensemble member the data examples are weighted differently which is claimed to lead to improved diversity.  In experiments, it is shown that calibration and model reliability is significantly improved over standard ensembles.

Overall, the submission received borderline scores in the initial reviewing round. Reviewer's requests for additional ablation studies, the effect of hyperparameters and comparisons to temperature scaling baselines have been mostly addressed by the rebuttal. Reviewers raised concerns about the connection to Bayesian posterior approximation to be ad-hoc, which were not fully clarified by the rebuttal. Moreover, there is a significant amount of literature in Bayesian deep learning with similar goals of improving calibration which has been missed by this work and more baselines could be compared to.

While many concerns were addressed by the author's rebuttal, the above open points make it difficult to accept the work for ICLR this time. For these reasons, the paper is not recommended for acceptance in its current form and I encourage the authors to take the reviewers feedback into account for a resubmission.

**Reviewer Concerns:**

- Lack of connection to Bayesian posterior approximation and overall ad-hoc formulation (Yz1U)
    - Remains mostly open, the connection of DRO to Bayes is not fully explained in the paper and rebuttal, and viewing ensembles as Bayesian posterior approximation remains heuristic.
- Limited experiments, small models only (Yz1U)
    - These were addressed by the rebuttal, 7B and 12B experiments added, method gives improvement over regular ensemble
- Missing comparison to temperature scaling (Yz1U)
    - Addressed by the rebuttal -- however, it seems that temperature scaling degrades ensemble performance
- Ablation study over individual components (AoME), impact of hyperparameters (AoME, CzTL), other small remarks (AoME)
    - All addressed by the rebuttal

**Reviewer Scores:**

Yz1U would have likely mainained the score, as I found the connection to Bayesian posterior approximation to be still unclear after reading the rebuttal.  AoME and CzTL may have raised their score, as their concerns are mostly addressed by rebuttal.

---

### Decision · Program_Chairs · 2026-01-26

Reject